# Diurnal evolution of total column and surface atmospheric ammonia in the megacity of Paris, France, during an intense springtime pollution episode

Rebecca D. Kutzner[1], Juan Cuesta[1], Pascale Chelin[1], Jean-Eudes Petit[2,3], Mokhtar Ray[1], Xavier Landsheere[1], Benoît Tournadre[1*], Jean-Charles Dupont[4], Amandine Rosso[5], Frank Hase[6], Johannes Orphal[6], and Matthias Beekmann[1]

[1] Laboratoire Interuniversitaire des Systèmes Atmosphériques (LISA), UMR CNRS 7583, Université Paris-Est
Créteil, Université de Paris, Institut Pierre Simon Laplace (IPSL), Créteil, France

[2] Laboratoire des Sciences du Climat et de l'Environnement, UMR 8212, CEA/Orme des Merisiers, 91191 Gif-sur-Yvette, France

[3] INERIS, Parc Technologique ALATA, 60750 Verneuil-en-Halatte, France

[4] Institut Pierre-Simon Laplace, École Polytechnique, UVSQ, Université Paris-Saclay, 91128 Palaiseau, France

[5] AIRPARIF, Agence de surveillance de la qualité de l'air, Paris, France

[6] Institut für Meteorologie und Klimaforschung (IMK), Karlsruher Institut für Technologie (KIT), Karlsruhe, Germany

*New affiliation: Centre for Observation, Impacts, Energy, Mines ParisTech, Sophia Antipolis Cedex, France

*Correspondence to*: Rebecca D. Kutzner (rebecca.kutzner@lisa.u-pec.fr) and Pascale Chelin (pascale.chelin@lisa.u-pec.fr)

**Abstract.** Ammonia ($NH_3$) is a key precursor for the formation of atmospheric secondary inorganic particles, such as ammonium nitrate and sulfate. Although the chemical processes associated with the gas-to-particle conversion
are well known, atmospheric concentrations of gaseous ammonia are still scarcely characterized. This information is however critical especially for processes concerning the equilibrium between ammonia and ammonium nitrate, due to the semi-volatile character of the latter one. This study presents an analysis of the diurnal cycle of atmospheric ammonia during a pollution event over the Paris megacity region in spring 2012 (five days in late March 2012). Our objective is to analyze the link between the diurnal evolution of surface $NH_3$ concentrations
and its integrated column abundance, meteorological variables and relevant chemical species involved in gas/particle partitioning. For this, we implement an original approach based on the combined use of surface and total column ammonia measurements. These last ones are derived from ground-based remote sensing measurements performed by the "Observations of the Atmosphere by Solar Infrared Spectroscopy" (OASIS) Fourier transform infrared observatory at an urban site over the southeastern suburbs of the Paris megacity. This
analysis considers the following meteorological variables and processes relevant to the ammonia pollution event: temperature, relative humidity, wind speed and direction and the atmospheric boundary layer height as indicator of vertical dilution during its diurnal development. Moreover, we study the partitioning between ammonia and ammonium particles from concomitant measurements of total particulate matter (PM) and ammonium ($NH_4^+$)

concentrations at the surface. We identify the origin of the pollution event as local emissions at the beginning of the analyzed period and advection of pollution from the Benelux and west Germany by the end. Our results show a clearly different diurnal behavior of atmospheric ammonia concentrations at the surface and those vertically integrated over the total atmospheric column. Surface concentrations remain relatively stable during the day, while total column abundances show a minimum value in the morning and rise steadily to reach a relative maximum in the late afternoon during each day of the spring pollution event. These differences are mainly explained by vertical mixing within the boundary layer, provided that this last one is considered well mixed and therefore homogeneous in ammonia concentrations. This is suggested by ground-based measurements of vertical profiles of aerosol backscatter, used as tracer of the vertical distribution of pollutants in the atmospheric boundary layer. Indeed, the afternoon enhancement of ammonia clearly seen by OASIS for the whole atmospheric column is barely depicted by surface concentrations, as the latter are strongly affected by vertical dilution within the rising boundary layer. Moreover, the concomitant occurrence of a decrease of ammonium particle concentrations and an increase of gaseous ammonia abundance suggests the volatilization of particles for forming ammonia. Furthermore, surface observations may also suggest night-time formation of ammonium particles from gas-to-particle conversion, for relative humidity levels higher than the deliquescence point of ammonium nitrate.

## 1. Introduction

Ammonia ($NH_3$) is a harmful air pollutant that directly affects human health and also contributes to intense smog events through the neutralization of sulfuric and nitric acids for forming secondary aerosols such as ammonium sulfate (($NH_4)_2SO_4$) and nitrate ($NH_4NO_3$) (Behera et al., 2013, Seinfeld and Pandis, 2016, Elster et al., 2018). These particles can be transported over long distances and contribute to the degradation of air quality and impact different ecosystems. Through conversion into different forms of reactive nitrogen, further impacts of ammonia and ammonium particles are directly or indirectly linked to acidic precipitation, acidification, eutrophication and loss of biodiversity e.g. (Sutton et al., 2011; 2013; Krupa, 2003). Depending on atmospheric temperature (T), relative humidity (RH) and the pH of the particles, volatilization of ammonium nitrate particles may form gaseous ammonia e.g. (Seinfeld and Pandis, 2016; Weber et al., 2016; Guo et al., 2018).

The main source of $NH_3$ in Europe is the agricultural sector, with an average of 93 % of total ammonia emission estimated for 2018 (Pinterits et al., 2020). It is emitted by volatilization from fertilizer storage, livestock as well as manure and mineral nitrogen fertilizers applied on crops, as a function of temperature, humidity and pH of atmosphere and soil as well as wind speed e.g. (Sommer et al., 2004, Behera et al., 2013). Other emissions are associated with traffic and industry. In France, the dominant source of $NH_3$ is also attributed to the agricultural sector, with contributions between 94 % to 98 %, among which 50 % is due to nitrogen-based fertilizers as well as emissions from livestock (Génermont et al., 2018, Ramanantenasoa et al., 2018). In many regions of Africa, Inner Mongolia, South Siberia and South America, fires are another anthropogenic source of $NH_3$ (Behera et al., 2013). Natural sources are related to biological mechanisms in soils, plants and soil-plant interaction, as described in detail by Behera et al. (2013). At global scale, ammonia emissions are mainly attributed to agriculture, biomass burning and the energy sector, accounting in 2005 for 80.6 %, 11.0 % and 8.3 % respectively (Behera et al., 2013). The European Union (EU) addressed ammonia emission in the National Emission Ceilings Directive 2001/81/EC (NECD). Serrano et al. (2019) recently reviewed reduction efforts of nitrogen levels between 2001 and 2011,

finding a significant impact of ammonia emitted from agriculture on ecosystems. Exceedances of ammonia emissions compared to ceilings set for 2010 are still occurring (NEC Directive reporting status 2018). New

reduction goals for the period of 2020 to 2029 and a second period after 2030 are set for each European country in the DIRECTIVE (EU) 2016/2284.

Pollution events in urban areas directly impact human health and greatly reduce visibility e.g. Molina and Molina (2004). This recurrently occurs during springtime over the Paris megacity (12.2 million inhabitants including suburbs) and other European megacities often associated with emissions from agricultural activities in the areas

surrounding the agglomerations e.g. Petit et al. (2015). Other pollution events in these areas are also linked to local or regional emissions of nitrogen oxides and sulfur dioxide from road traffic and industry (Behera and Sharma, 2010). Accurate and long-term measurements of atmospheric pollutants, such as ammonia, and meteorological conditions are crucial in order to better understand the origin and the evolution of these pollution events. In the Paris region, springtime is a very propitious period for particulate matter pollution episodes,

essentially dominated by secondary inorganic aerosols, such as ammonium nitrate and sulfate (Sciare et al., 2011, Petit et al., 2015). Concomitantly, ammonia concentrations have been found exceptionally high, as reported by surface in situ measurements (Petit et al., 2015; Petetin et al., 2016) and remote sensing from the ground and satellite (Tournadre et al., 2020; Viatte et al., 2020). Indeed, that period of the year is characterized by fertilizer spreading, which can dramatically enhance $NH_3$ emissions (Ramanantenasoa et al., 2018).

Different techniques are used to measure concentrations of $NH_3$ in the atmosphere. Difficulties to measure ammonia by in situ techniques are associated with its "sticky" nature, inducing its accumulation in inlets and sampling tubes. In order to reduce these artefacts, different techniques are often implemented, such as the use of polyethylene or Teflon tubes (instead of steel or silicosteel), halocarbon wax coating, while keeping the length of the tubes to a minimum possible and a heating system for reducing relative humidity that may also lead to losses

of $NH_3$ (Yokelson et al., 2003, Whitehead et al., 2008).

Remote sensing of ammonia is an innovative alternative to in situ techniques, which offers a significant enhancement of spatial coverage. Satellite approaches are currently based on hyper spectral thermal infrared measurements from the Cross-track Infrared Sounder (CrIS, Shephard and Cady-Pereira, 2015) and the Infrared Atmospheric Sounding Interferometer (IASI, Clerbaux et al., 2009), respectively onboard the United States Suomi

National Polar Partnership (SNPP) and the European MetOp satellites. Both platforms are pointing nadir in polar sun-synchronous orbits, with overpasses around 09:30 and 21:30 local time (LT) for IASI and 13:30 and 01:30 LT for CrIS (Shephard and Cady-Pereira, 2015, Dammers et al., 2017). Therefore, they both offer global coverage twice a day, providing particularly valuables measurements over remote regions lacking of ground-based instruments such as in the tropics. Remote sensing of ammonia can also be performed using hyperspectral

measurements from a ground-based Fourier-transform infrared (FTIR) spectrometer, like the OASIS (Observations of Atmosphere by Solar Infrared Spectroscopy, Chelin et al., 2014) mid-resolution observatory in Créteil (France). Remote sensing from satellite and ground-based platforms provides vertically integrated amounts of ammonia over the atmospheric column for cloud-free conditions. The combined use of remote sensing and in situ measurements offers an interesting framework for analyzing ammonia variability both at the surface and

integrated over the atmospheric column, as already done for greenhouse gases (Zhou et al., 2018). FTIR ground-based measurements can provide highly valuable information on the diurnal evolution of atmospheric species for a particular geographical location, as shown here for ammonia at the Paris suburbs. Although numerous FTIR

ground-based stations currently exist, such as those of the NDACC network (De Mazière et al., 2018), only few of them document the diurnal evolution of atmospheric constituents.


This paper presents a detailed analysis of the diurnal evolution of ammonia as observed in total columns from ground-based remote sensing and at the surface from an in-situ analyzer in Paris, France, during a major pollution event in late March 2012. We characterize the diurnal variation of ammonia, analyzing both the link with the formation and volatilization of ammonium particles and vertical dilution in the atmospheric boundary layer.

Spring 2012 was one of the most polluted periods since 2007, with a succession of persistent pollution events (Petit et al., 2015, Petit et al., 2017). We use total column ammonia concentrations derived from OASIS observatory at southeast Paris suburbs (Créteil) and surface observations at southwest Paris suburbs (Palaiseau) to characterize the diurnal evolution of ammonia between 26 and 30 March 2012. To the authors' knowledge, this is the first analysis of the diurnal evolution of ammonia from both total column and surface measurements, in

close relation with particle phase measurements.

Section 2 provides information concerning the instruments from the OASIS and SIRTA sites, as well as other datasets used for this study. We also provide a brief description of the new retrieval of ammonia from OASIS (Tournadre et al., 2020). In the third section, we present and discuss the analysis of these datasets. First, we

describe the regional conditions of the Paris pollution event in late March 2012 using meteorological analysis, a chemistry-transport model and satellite data (Sect. 3.1). Then, we analyze the diurnal evolution of surface and total columns of ammonia and particulate matter as well as meteorological variables over the Paris region (Sect. 3.2 to 3.4). Following, we analyze the complementarity of surface and total column measurements of ammonia using ground-based backscatter lidar (LIght Detection And Ranging) measurements, as proxy for the vertical

distribution of pollutants within the atmospheric boundary layer (Sect. 3.5). Section 4 provides conclusions of this study.

## 2. Datasets

### 2.1 Description of ground-based sites and platforms

An original aspect of this work is the analysis of the diurnal evolution of total column observations of ammonia

derived from OASIS. This remote sensing observatory is located in Créteil (OASIS; 48.79° N, 2.44° E; 56 m above sea level (asl)), at the southeast suburbs of Paris, on the rooftop of the Université Paris-Est Créteil (UPEC, Chelin et al., 2014). It is an urban site mainly affected by background levels of pollution (Figure 1).

Measurements from other sites over the Paris region are also used in the current study (Figure 1). Meteorological and detailed atmospheric composition data at the surface level are measured at the "Site Instrumental de Recherche

par Télédétection Atmosphérique" supersite near Palaiseau (SIRTA; 48.72° N, 2.20° E; http://sirta.ipsl.fr), located about 19 km southwest from OASIS and southwest Paris, which is often used for monitoring background air quality conditions of the Paris region (Haeffelin et al., 2005). We use radiosounding measurements of temperature, pressure and humidity profiles from the Trappes station about 31 km west of Créteil, 15 km away from the SIRTA supersite and operated by Météo France. Additional surface measurements of $PM_{2.5}$ and $PM_{10}$ (particle matter

with aerodynamic diameter respectively less than 2.5 and 10 µm) are provided by the Airparif network dedicated

to monitoring air quality in the Paris region (https://www.airparif.asso.fr/) from the stations of Vitry-sur-Seine, Bobigny and Gennevilliers. In the paper, time series of measurements are presented in terms of hourly median values, except stated otherwise.

**2.2 Observations of total column ammonia derived from OASIS**

Since 2009, the OASIS observatory regularly records high spectral measurements of solar radiation absorbed and scattered by atmospheric constituents, under clear-sky conditions (Chelin et al., 2014). It uses a mid-spectral resolution Fourier-transform spectrometer manufactured by Bruker Optics (the Vertex 80 model) with a spectral resolution of 0.06 cm$^{-1}$ (corresponding to a maximum optical path difference of 12 cm). OASIS is routinely used for monitoring air pollutants, such as tropospheric ozone ($O_3$) and carbon monoxide (CO), with good accuracy

and high sensitivity to near surface concentrations (Viatte et al., 2011). This system is particularly suited for air quality monitoring in megacities, given its compactness and moderate cost and it can play a key role in validating current (e.g. IASI) and future satellite observations (e.g. Infrared Atmospheric Sounder Interferometer Next Generation - IASI–NG and the InfraRed Sounder onboard the Meteosat Third Generation mission - MTG/IRS). The observatory is covered by an automatized cupola (Sirius 3.5 "School Model" observatory, 3.25 m high and

3.5 m in diameter), in which the aperture rotates for tracking the solar position. The altitude-azimutal solar tracker of OASIS uses bare gold-coated mirrors (A547N model from Bruker Optics). Infrared solar radiation spectra are recorded by a DTGS (deuterated triglycine sulphate) detector using a potassium bromide (KBr) beamsplitter to cover the large spectral region from 700 to 11000 cm$^{-1}$ (0.9–14.3 μm) with no optical filter. The acquisition system is set to average over 30 scans at maximum spectral resolution in order to increase the signal-to-noise ratio of the

measurements. This averaging procedure results in an effective temporal resolution of 10 minutes, that allows measuring the diurnal variability of relatively short-lifetime species such as $NH_3$. Absolute calibration of spectra measured by OASIS is done every month with a reference internal source of radiation.

Ammonia concentrations integrated over the total atmospheric column are retrieved with the PROFFIT 9.6 code developed by the Karlsruhe Institute of Technology (Germany, Hase et al., 2004), adapted for the medium spectral

resolution. As detailed by Tournadre et al. (2020), two spectral micro-windows within the $\nu_2$ vibrational band of $NH_3$ are used: 926.3-933.9 cm$^{-1}$ and 962.5-970 cm$^{-1}$. The main interfering species in this spectral range are water vapour ($H_2O$), carbon dioxide and $O_3$, whose abundances are taken from the Whole Atmosphere Community Climate Model (WACCM version 6: Chang et al., 2008) and jointly adjusted with that of $NH_3$. We also use climatological concentrations for minor interfering gases (i.e. nitric acid $HNO_3$, sulphur hexafluoride $SF_6$, ethane

$C_2H_4$, and chlorofluorocarbons - CFC-12) that may essentially impact the baseline of the spectra. The spectral signatures of absorption of infrared radiation by ammonia is clearly seen in individual spectra measured by OASIS, such as those recorded during a pollution event during March 2012 (as compared to the atlas from Meier et al. (2004), see Fig. 2). Atmospheric columns of ammonia derived from the 9 year-database of OASIS range from 0.0005 $10^{16}$ to 9 $10^{16}$ molecules per square centimetre (molec.cm$^{-2}$) and their retrieval error is estimated to

20-35% (Tournadre et al., 2020), dominated by the systematic errors that are the combination of uncertainties in the spectroscopic parameters of ammonia and the interfering species (the dominating term), radiometric noise, instrumental parameters, and forward model uncertainties. The magnitude of these errors are comparable to those estimated by Dammers et al. (2015) for a high-resolution ground-based station at Bremen (Germany). OASIS retrievals of $NH_3$ total columns show a good agreement with co-located observations derived from IASI (the

ANNI-NH3-v2.2R version, Van Damme et al., 2017): a linear correlation coefficient of ~0.8 and a small mean

difference of ~0.08 $10^{16}$ molec.$cm^{-2}$, with OASIS-derived concentrations slightly larger (Tournadre et al., 2020).

This last aspect could be associated with an enhanced sensitivity to larger concentrations of $NH_3$ near the surface

for OASIS, as compared to the satellite retrieval which is most sensitive for higher atmospheric layers.

**2.3 Surface in situ observations of ammonia and aerosol composition**

In the present analysis, we use in situ gaseous ammonia measurements at surface level carried out with an

AiRRmonia instrument (Mechatronics Instruments, the Netherlands) at the SIRTA observatory (Haeffelin et al.,

2005). The principle of this instrument, described in Cowen et al. (2004), is essentially based on conductimetric

detection of ammonia that is first absorbed via a gas-permeable membrane and dissolved in water (i.e. in the form

of ammonium ions, forming acidic solution). Several intercomparison exercises have shown that this procedure

provides more accurate $NH_3$ measurements (Norman et al., 2009, von Bobrutzki et al., 2010). The AiRRmonia

was regularly calibrated with 0 and 500 ppb ammonium solution.

Concomitant measurements of the major chemical composition of submicron aerosols were performed with an

Aerosol Chemical Speciation Monitor (ACSM, Aerodyne Research Inc., Billerica, MA, USA; Ng et al., 2011),

providing concentrations of particulate organic matter (OM), nitrate ($NO_3^-$), sulphate ($SO_4^{2-}$), ammonium ($NH_4^+$)

and chloride ($Cl^-$), every 30 minutes. Submicron particles are sampled at 3 l $min^{-1}$, subsampled at 0.85 l $min^{-1}$, and

focused through an aerodynamic lens for $PM_1$ (particle matter with aerodynamic diameter smaller than 1 μm).

Non-refractory particles are then flash-vaporized on a 600°C-heated plate, fragmented by electronic impact at 70

keV, and eventually separated and detected by a quadrupole. Calibrations were performed by injecting known

concentrations of ammonium nitrate and ammonium sulfate particles with an aerodynamic diameter of 300 nm.

Details on the operational conditions of the AiRRmonia and the ACSM instruments at SIRTA are provided by

Petit et al. (2015).

As observed by Petit et al. (2015), we expect the daily evolution of ammonia over the Paris region to be closely

linked to the gas to particle conversion between ammonia (gas) and ammonium nitrate particles. This is a

reversible conversion for which the equilibrium is closely linked to the abundance of precursors ($NH_3$ and $HNO_3$),

and meteorological conditions, temperature and relative humidity (Seinfeld and Pandis, 2016). The conditions

needed for volatilization of $NH_3$ from $NH_4NO_3$ are given by the relationship of relative humidity and

deliquescence relative humidity (DRH), which depends on temperature. Whereby volatilization is favored when

RH is much lower than DRH. In order to estimate the balance between DRH and RH, we consider the following

equation, as suggested by Seinfeld and Pandis (2016):

$$DRH(T) = DRH(298)exp\left\{\frac{\Delta H_S}{R}\left[A\left(\frac{1}{T}-\frac{1}{298}\right) - B\ ln\frac{T}{298} - C(T-298)\right]\right\}$$

DRH(298) is the deliquescence relative humidity of $NH_4NO_3$ at 298 K, which corresponds to 61.8 %. $\Delta H_S$ is the

enthalpy of solution for $NH_4NO_3$ at 298 K which is 25.69 kJ $mol^{-1}$, R is the universal gas constant, T is the

temperature in K. A, B, and C are factors for the solubility of common aerosol salts in water as a function of

temperature provided by Seinfeld and Pandis (2016) (i.e. 4.3, -3.6 $10^{-2}$ and 7.9 $10^{-5}$ respectively). Moreover,

partitioning between ammonia and ammonium nitrate is also influenced by the pH of the ambient particles (e.g.

Weber et al., 2016; Guo et al., 2018). When pH drops below an approximate critical value of 3 (slightly higher in

warm and slightly lower in cold seasons), the $NH_3$ reduction leads to evaporation of $NH_4NO_3$, while this is not

expected to happen for moderately acid to neutral conditions (Guo et al., 2018). In addition, it is worth noting that in the present study we use the above expression and the currently available data for a qualitative interpretation of diurnal variations of ammonia and ammonium. However, additional dedicated measurements throughout the atmospheric column are needed in order to perform a quantitative analysis (see more details in the conclusion section).

### 2.4 Regional conditions from satellite data and models

For characterizing the pollution event during March 2012, we use a suite of satellite and model datasets concerning both the pollutant distributions at regional and continental scale and meteorological conditions. Aerosol optical depth (AOD) derived from satellite and ground-based measurements are used for analyzing the spatial and temporal evolution of total particle abundance integrated over the atmospheric column. The horizontal distribution of AOD over western Europe is described using MODIS (Moderate Resolution Imaging Spectroradiometer, Remer et al., 2005) data onboard the Terra (MOD04L2) satellite with overpasses at 10:30 LT (from the NASA worldview website https://worldview.earthdata.nasa.gov/; Levy, R., and Hsu, C., 2015). The MODIS images have a horizontal resolution of 3 km at nadir.

The horizontal distribution of air pollutants at the European scale is studied with CHIMERE chemistry-transport model simulations of $PM_{2.5}$ provided by the ESMERALDA (EtudeS Multi RégionALes De l'Atmosphère, Cortinovis et al., 2015) project (http://www.esmeralda-web.fr/accueil/index.php). The version 2008b of CHIMERE is run hourly and averaged at daily scale, with a horizontal resolution of 15 x 15 $km^2$ and 9 vertical levels between 20 m to 5 km. Meteorological inputs for CHIMERE come from MM5 simulations (Dudhia, 1993), using Final Analyses (FNL) data from National Centers for Environmental Prediction (NCEP) as boundary conditions. Chemical reactions are simulated using the MELCHIOR2 mechanisms scheme and the ISORROPIA model (Nenes et al., 1998). This last one has been used to produce tables that are inserted in the CHIMERE model for calculating the thermodynamic equilibrium of the species. Ammonia, nitrate and sulfate are simulated in aqueous, gaseous and particulate phases in the model.

Meteorological conditions are analyzed from in-situ measurements and numerical model simulations. We use sea-level pressure, wind and potential temperature fields from ERA-Interim (ERAI, Simmons et al., 2007) reanalyzes of the European Centre for Medium-Range Weather Forecasts (ECMWF) that are provided by the Institut Pierre Simon Laplace mesocentre (https://mesocentre.ipsl.fr). These simulations have a 0.75 x 0.75° horizontal resolution and 37 pressure levels.

### 2.5 Local conditions at the Paris region from ground-based measurements

Meteorological conditions at the surface over the Paris region are analyzed by in situ measurements of wind speed and direction performed at the SIRTA site (Haeffelin et al., 2005). Local temperature and relative humidity were measured at Créteil with a LOG 110-EXF sensor, with an accuracy in temperature of ± 0.5°C and in relative humidity of ± 3%.

Vertical profiles of temperature and relative humidity from the surface up to 25 km of altitude and with about 10-m vertical resolution are measured by radiosoundings launched around noon and mid-night at the Trappes site (at southwest suburbs of Paris).

The diurnal evolution of particle pollution over the Paris region is studied in terms of surface measurements of $PM_{2.5}$ and $PM_{10}$ from several Airparif sites and AOD measured by ground-based sun photometers (version 3 of

level 2.0 data) at the Paris and SIRTA sites from the AERONET (Aerosols Robotic Network, Holben et al., 2001, https://aeronet.gsfc.nasa.gov/) network. We use the distinction between AOD from a fine (e.g. smoke or smog) and coarse (e.g. sea-salt or dust) mode at 500 nm, derived from the wavelength dependence of the AOD (O'Neill et al., 2003; Giles et al., 2019). Errors in AOD data correspond to approximately 0.02 (Giles et al., 2019). Additionally, we use ground-based lidar measurements from the SIRTA site for describing the vertical distribution

of particles over the Paris region, which is used as an indicator of the vertical distribution of air pollutants and the vertical structure of the atmospheric boundary layer. This is done with vertical profiles of attenuated backscatter profiles, measured by an elastic backscatter lidar (the ALS model manufactured by Leosphere) at 355 nm. The mixing boundary layer height is visually identified as the lowest marked discontinuity of the lidar profiles during daytime hours (from 06:00 to 18:00 UTC).


## 3. Results

We focus our study on the diurnal evolution of ammonia during a major pollution event over the Paris region occurring at the end of March 2012. It corresponds to the period with highest concentrations of ammonia on the multiyear time series (2009–2017) of OASIS measurements, which is probably linked to the springtime spreading

of mineral fertilizer in the Paris region and the surrounding regions (Ramanantenasoa et al., 2018; Tournadre et al., 2020). It is the most polluted spring season between 2007 and 2015 (Petit et al., 2017).

### 3.1. Meteorological and atmospheric conditions over western Europe

During late March 2012, the prevailing atmospheric conditions over western Europe are driven by an anticyclonic high-pressure system centered over Great Britain and the North Sea (55°N, 0°E) on 26 March and moving

westwards in the following days (see Fig. 3a, d, g). Following the anticyclonic circulation associated with this system, north-easterly winds blow from the Benelux region (Belgium, Netherlands and Luxembourg) to northern France. As expected for an anticyclonic period, relatively low wind speeds occur at its core, located over central Europe (from southern France to eastern Germany), which are accompanied by high insulation and low cloudiness (not shown). According to MODIS satellite observations (Fig. 3b, e, h) and CHIMERE simulations (Fig. 3c, f, i),

an aerosol plume with moderate AOD (0.2 to 0.3) and moderately large concentrations of $PM_{2.5}$ at the surface (20 to 30 µg m$^{-3}$) is formed on 26 March over the Benelux and extends across the English Channel. Meanwhile, aerosol baseline levels are observed over northern France (AOD ~0.1 and 10-15 µg m$^{-3}$ for surface $PM_{2.5}$). After

27 March, the aerosol plume reaches northern France and southern England. On 28 March, a clear enhancement of the aerosol load over the Benelux and northern France is observed both in terms of AOD (up to 0.4) and

305 modeled surface $PM_{2.5}$ concentrations (up to 50 µg m$^{-3}$). These high aerosol loads over northern France remain until 30 March (not shown). Both the horizontal extent of the aerosol plume and wind directions suggest that these

highly polluted air masses originate over the Benelux as well as west Germany and are transported southwestwards, clearly reaching the Paris region after 28 March (also remarked for this pollution event by Fortems-Cheiney et al., 2016).

## 3.2 Geographical distribution of particle matter over the Paris region

Over the Paris region, particle concentrations at the surface are moderately high on 26-27 March ($PM_{2.5}$ concentrations up to 40 µg m$^{-3}$, see surface measurements of PM levels from several stations of the Paris region in Fig. 4). As polluted air masses are advected from the Benelux and west Germany on 28-29 March, $PM_{2.5}$ levels are clearly enhanced (up to 80 µg m$^{-3}$), as also seen in daily averaged simulations. Figure 4 also shows the largest peaks of surface $PM_{2.5}$ concentrations occurring every day during the morning and secondary high values in the late evening.

Very similar temporal evolutions of surface particle concentrations are observed over the whole Paris region and during the entire period (26-30 March), both in absolute and relative terms. Figure 4 illustrates this horizontally homogeneous distribution of surface PM as the chosen stations are located at the southeast, southwest, northeast and northwest suburbs of Paris (Fig. 1). The same peaks and troughs of surface PM are seen for all these locations. Particularly, we also remark that $PM_1$ at SIRTA also shows the same temporal evolution as other stations in the Paris region, but with levels roughly ∼30 % below those of $PM_{2.5}$ on 26-28 March and similar concentrations afterwards (for both $PM_1$ and $PM_{2.5}$). The 30 % difference between $PM_1$ and $PM_{2.5}$ observed in the present case could be linked with aging (and/or long-range transport) which has been remarked for measurements in 2015 by Petit et al., (2017). In the Paris region, the $PM_1$ generally represent 90% of $PM_{2.5}$ (Petit et al., 2017), particularly when $PM_1$ is larger than 20 µg m$^{-3}$ (although for lower levels, $PM_1$ may represent around 50% of PM2.5, Petit (2014)). Occasionally, some background levels of PM might not be accounted in $PM_1$ that are measured as $PM_{2.5}$ (Petit, 2014). Moreover, comparisons made by Petit (2014) show a very similar statistical distribution for both $PM_1$ at SIRTA and PM2.5 at the urban background stations at Paris suburbs mentioned in Fig. 4. For the future, it should be very interesting to have collocated $PM_1$ and $PM_{2.5}$ chemical composition measurements. The clear similarity of these measurements at four different locations of the Paris suburbs suggests that we may also expect a consistent evolution of pollution levels at the Créteil site (OASIS observatory), whose observations are also used later in this section for analyzing the evolution of the atmospheric ammonia concentrations during the event.

The time series of surface PM levels suggest the occurrence of 2 distinct pollution regimes within the period of 26-30 March. Indeed, while daily mean $PM_{2.5}$ values on 26-27 March remain under the air quality 24 h guideline of WHO (World Health Organization, $PM_{2.5}$ of 25 µg m$^{-3}$, except for one station on one day), this $PM_{2.5}$ threshold is exceeded for all stations on 28-30 March. Hereafter, these two regimes are named as: period 1 or P1 (26-27 March) and period 2 or P2 (28-30 March). These two different atmospheric conditions are also pointed out by Petit et al. (2015) by analyzing this particular pollution episode using surface measurements at SIRTA. A statistical comparison of the similarity of surface PM measurements from different sites over the Paris region is shown in Table 1 (for periods 1 and 2). For the first period (26-27 March), the time series of hourly $PM_{2.5}$ measurements performed at three different locations show a moderate correlation between each other ($R^2$ of 0.63 to 0.67), suggesting a similar evolution but with some horizontal heterogeneity over the Paris region. During the second period (28-30 March), correlations between PM measurements are clearly higher ($R^2$ of 0.86 to 0.91) and therefore indicate a more horizontally homogenous PM distribution over the Paris region. Levels of surface $PM_{10}$

for the same stations and periods show similar behaviors (Table 1). This different behavior between P1 and P2 is likely linked to the origin of the pollution event, being rather local for P1 and dominant advection of air pollution from the Benelux during P2, as remarked in the regional analysis of AOD, PM and wind regimes of section 3.1. Additionally, we note that comparisons of $PM_{2.5}$ from 3 stations to $PM_1$ at SIRTA show moderate correlations during P1 as the other measurements, but lower ones during P2 (although peaks and troughs are clearly coincident).

**3.3 Evolution of ammonia concentrations over the Paris region**

During the periods P1 and P2, ammonia concentrations over the Paris region are observed both at surface level (using in situ analyzer at SIRTA) and integrated over the total atmospheric column (using OASIS at Créteil, Fig. 5). Total atmospheric columns of $NH_3$ show a very marked and clear diurnal evolution: lower column amounts of ammonia in the morning that rise almost monotonically during the day until reaching a maximum in the afternoon. Both on 26 and 27 March, stable ammonia concentrations around $2 \cdot 10^{16}$ molec.cm$^{-2}$ remain until noon and then increase only in the afternoon. Early morning total columns of $NH_3$ on 28 and 29 March are lower (respectively $1.4 \cdot 10^{16}$ and $0.6 \cdot 10^{16}$ molec.cm$^{-2}$) than for the previous days and show a steady enhancement from the early morning to the afternoon. The highest total column of ammonia is measured on 28 March ($4.6 \cdot 10^{16}$ molec.cm$^{-2}$). On 30 March, the diurnal evolution of $NH_3$ total columns is more similar to the first two days (26-27 March). Steady total columns around $1.5 \cdot 10^{16}$ molec.cm$^{-2}$ during the first 1.5 hours of the morning are followed by a decrease down to $0.85 \cdot 10^{16}$ molec.cm$^{-2}$ around 11:00 UTC and afterwards an increase up to $2.5 \cdot 10^{16}$ molec.cm$^{-2}$, which is lower than those observed during the previous 4 days. This clear enhancement of ammonia total atmospheric columns during the day measured by OASIS is found typical of springtime polluted periods as already analyzed by Tournade et al. (2020) but not showed here (e.g. in March 2014, and March 2016). For all these years, the $NH_3$ maximum in the afternoon is above $2 \cdot 10^{16}$ molec.cm$^{-2}$ (Tournadre et al., 2020).

Meanwhile, surface measurements at SIRTA show relatively high overall levels of ammonia: from 2 to 10 µg m$^{-3}$, which is higher than Paris urban background levels of 1-3 µg m$^{-3}$ shown by Petetin et al. (2016), but for a May 2010 to February 2011 period. On each of the days of the event (26-30 March), morning daily maxima (up to 6-9 µg m$^{-3}$) and smaller evening peaks (around 5 µg m$^{-3}$, Fig. 5) are clearly depicted. Although both surface (Fig. 5b) and integrated total column (Fig. 5a) ammonia measurements show large concentrations, their daily evolutions are clearly different. While total column values increase steadily during the day until reaching a peak in the late afternoon, surface ammonia moderately fluctuates during the day. These differences may be associated with atmospheric processes or interactions with the surface that modify ammonia concentrations differently as a function of altitude. This may be the case for vertical dilution of atmospheric constituents within the atmospheric boundary layer or the vertical variability of gas/particle partitioning related to relative humidity, temperature and particle pH. These aspects are investigated in detail in the following paragraphs.

Vertical variations of atmospheric ammonia concentrations may potentially be associated with temperature and relative humidity vertical profiles. As mentioned in Sect. 2.3, dry conditions lead to volatilization of ammonia from ammonium particles, whereas humidity levels beyond the deliquescence point favor the inverse process (Seinfeld and Pandis, 2016). During the pollution event on 26-30 March, temperature shows the usual steady decrease with altitude from the surface up to 2.5 km (see the median temperature profile measured by radiosoundings launched at Trappes on 26-30 March, Fig. 6a). Relative humidity varies greatly at the lowest few

kilometers of the atmosphere, typically increasing with altitude within the mixing boundary layer (up to 1 to 1.5 km above sea level, asl, for the present case, see Fig. 6b). On 26-27 and 29 March, relative humidity increases from 25 % at the surface up to 35-40 % around 900 m asl and drops above 1000 m asl down to 10-20 %. On 28 March, relative humidity is roughly 15 % higher than on the mentioned days up to 800 m asl, above which it decreases down to 48 % and then rises up to 60 % at 1600 m asl, dropping down to 30 % at 2500 m asl. In all

these cases, relative humidity up to 2500 m asl remains below the deliquescence point (DRH) as shown in Fig. 6b, thus favouring the formation of $NH_3$ by volatilization of ammonium particles. This is also confirmed by relative humidity time series at different altitudes (200, 500 and 1000 m asl), reconstructed from all radiosounding measurements over the entire event (launched from Trappes both at mid-day and mid-night, Fig. 6c). This supports a hypothesis of an increase in ammonia amounts due to volatilization of ammonium nitrate at higher altitudes.

Relative humidity always remains below the DRH (grey band), except for one single measurement at 1000 m on 30 March at noon. We also remark that during the whole period relative humidity does not vary much vertically below 1000 m (except on 30 March) and that the most humid conditions are found at mid-nights from 28 to 30 March. No contrasting conditions between mid-day and mid-night are either found for the vertical shape of relative humidity. We do not clearly point out any particular link or concomitant temporal variation of relative humidity

every 12 h at different altitudes (Fig. 6c) and ammonia measurements (Fig. 5).

    An additional analysis was performed with the ISORROPIA II box model (Fountoukis and Nenes, 2007) to investigate the role of temperature and relative humidity in the partitioning of ammonium nitrate. The forward calculation used measurements of the SIRTA site for $NH_4^+$, $NO_3^-$ and $NH_3$ on 28 March 2012 representing the highest concentrations on the studied period, as well as the meteorological parameters. $HNO_3$ concentrations were

set constant from values in Petetin et al., (2016) for the same period of the year. As expected, results indicate that partitioning of ammonia into the particulate phase is favored with the decrease of temperature. This temperature decrease is correlated to an increase of relative humidity (while values remaining below DRH). Therefore, in equilibrium conditions (e.g. in absence of ammonia and ammonium advection), ammonia likely decreases at higher altitudes. Nevertheless, it should be noted that pH and aerosol chemical composition also impact

ammonium nitrate partitioning. $PM_1$ was found to be neutralized during the period study (Figure S1 in the supplementary material), therefore we expect a limited influence of ambient particle pH. However, our full understanding is limited by the lack of $HNO_3$ in situ and column measurements.

    As a conclusion, the decrease in T and the increase of RH within the boundary layer height of 1-1.5 km with respect to ground shift the equilibrium to the aerosol phase. This does not explain the observed day time columns

$NH_3$ maximum, which was not observed at the surface. This suggests that T and RH may not be the only driver regarding the vertical variability in $NH_3$ concentrations. Other possible drivers are analysed in the coming sections.

### 3.4 Link between ammonia and ammonium particles over the Paris region

    A joint analysis of the temporal evolution of ammonia and ammonium particles provides further evidence on the

role of particle/gas conversion on the evolution of ammonia concentrations. As previously mentioned, volatilization leads to concomitant increases of ammonia concentrations and decreases of $NH_4NO_3$ particles (the most abundant ammonium particles observed during this event, Petit et al., 2015). Complementary, the formation of ammonium nitrate particles may be accompanied by a relative reduction of the abundance of its precursors (if

they are not in excess), thus ammonia and HNO$_3$. The following two subsections analyze these processes for the periods 1 and 2.

### 3.4.1 Local pollution regime on 26-27 March 2012 (period 1)

Figure 7 presents hourly median measurements of ammonia total column from OASIS and surface concentrations of ammonia, ammonium (NH$_4^+$), nitrate (NO$_3^-$) and sulfate (SO$_4^{2-}$) radicals measured at the SIRTA site (respectively in Figs 7a, c, e, for P1 and 7b, d, f, for P2). During P1, hourly ammonia total columns measured by OASIS show a stable level around 2 10$^{16}$ molec.cm$^{-2}$ until 10:00 UTC, after which a steady increase with larger variability is observed during all the afternoon until reaching a median maximum of 3.4 10$^{16}$ molec.cm$^{-2}$ around 15:00 UTC (Figure 7a). Surface ammonia concentrations strongly vary during the night and clearly increase in the morning hours with a relative maximum around 07:00 UTC up to 7 μg m$^{-3}$, likely related to evaporation from morning dew (Petit et al., 2015; Wentworth et al., 2016) and it is followed by a steady decrease of about 35 % down to 4.5 μg m$^{-3}$ around 13:00 UTC. A second relative enhancement of surface NH$_3$ is seen during the afternoon at 17:00 UTC until reaching 6 μg m$^{-3}$, after which it fluctuates with concentrations around 5 μg m$^{-3}$ until midnight (Figure 7c). Hourly concentrations of NH$_4^+$ and NO$_3^-$ show a similar evolution during the day (Figs. 7c, e). They remain rather stable during the night and early morning hours until 06:00 UTC (around ~5 and ~18 μg m$^{-3}$ for respectively NH$_4^+$ and NO$_3^-$), with a relative peak at 03:00 UTC. Afterwards, their concentrations show a small relative peak at 07:00 UTC, followed by a strong reduction of 75 % during the day. Between noon and 19:00 UTC, a rather stable daily minimum is seen for both ammonium and nitrate concentrations (of respectively ~1 and ~3 μg m$^{-3}$). This is followed by a slightly increase (up to 3 and 7 μg m$^{-3}$ respectively). Meanwhile, sulfate amounts remain low during P1 (below 1.5 μg m$^{-3}$, Fig. 7e).

From the early morning until the afternoon, the strong reduction of 75 % for both ammonium and nitrate abundances at the surface, which is not clearly observed for surface ammonia (reducing by only 35 %), likely suggest the occurrence of volatilization of ammonium particles. Probably, particle volatilization may be observable locally at the Paris region as this first period (P1) is characterized by a rather local pollution regime with limited transport of pollutants from other regions. Sustained volatilization of ammonium particles would lead to a steady enhancement of ammonia concentrations, as it is clearly observed during almost all daytime by OASIS in terms of NH$_3$ total columns. Additionally, volatilization of applied mineral fertilizers in the surrounding crop areas may also contribute to the daytime increase of ammonia, as analyzed in detail during the same period (March/April 2012) over crop fields located west of Paris by Personne et al. (2015).

The fact that the daytime enhancement of NH$_3$ is not that clearly reflected by its variability at surface level (Fig. 7c) might be associated with an additional phenomenon that would reduce surface concentrations of gases and particles during daytime, such as vertical mixing within the atmospheric boundary layer (which is investigated in detail in Sect. 3.5). Volatilization of ammonium particles is also favored by rather dry conditions during the day, with surface relative humidity dropping down to 19 % at about 12:00 UTC, while the deliquescence relative humidity point is 65 % (Fig. 8c). Local meteorological conditions are also characterized by a gentle to fresh breeze, according to the Beaufort scale, with a dominant wind direction from the north (see Figs. 8e and 8g) and surface temperatures ranging from 11 to 21°C.

### 3.4.2 Pollution transported from the Benelux region and west Germany on 28-30 March 2012 (period 2)

The P2 is characterized by the arrival of polluted air masses to the Paris region, originating from the Benelux and west Germany region (as remarked in Sect. 3.1, Fig. 3c, f, i) by rather weak winds (2.5 m s$^{-1}$, Fig. 8f) from the north and northeast (Fig. 8h). Locally, meteorological conditions favor particle formation during the night (maximum of RH of 95 % above a DRH of 67 %, Fig. 8d) and volatilization during the day (minimum of RH of 27 % well below the DRH of 66 %). A clear and stronger afternoon enhancement of ammonia total columns is observed (Fig. 7b) as compared to the two previous days (P1, Fig. 7a). The median diurnal evolution of ammonia columns during P2 depict an early decrease from 1.5 10$^{16}$ molec.cm$^{-2}$ at 07:00 UTC down to 1 10$^{16}$ molec.cm$^{-2}$ at 10:00 UTC. Then, they steadily rise for 6 hours until reaching 3.5 10$^{16}$ molec.cm$^{-2}$ at 16:00 UTC (Fig. 7b) with clearly more variability than during P1. Surface measurements of NH$_3$ during the night show a steady decrease from 5 μg m$^{-3}$ at midnight to 3 μg m$^{-3}$ at 05:00 UTC, with smaller variability as compared to the same period of the day during P1. Surface NH$_3$ concentrations increase during the morning until reaching a maximum of 5.7 μg m$^{-3}$ at 08:00 UTC, followed by a reduction down to 4 μg m$^{-3}$ two hours after (10:00 UTC) and then a second relative maximum of smaller amplitude (5 μg m$^{-3}$) in the afternoon (15:00 UTC, Fig. 7d). Ammonia emissions from the soil of surrounding crop areas may contribute to its enhancement during the morning (Personne et al., 2015). It is also worth noting that forests surrounding the NH$_3$ surface measurement site at Palaiseau may act as local sinks of ammonia (as remarked by Behera et al., 2013, Hansen et al., 2015), this is not the case for total column retrievals performed at Créteil.

When it comes to the particle components at surface level, all inorganics (ammonium, nitrate and sulfate) exhibit relatively large amounts and follow similar diurnal evolutions, which is probably associated with the arrival of air pollutants rich in nitrate (Fig. 7f; also remarked by Petit et al., 2015) from the Benelux and west Germany (Fortems-Cheiney et al., 2016). The concentrations of these three particle components steadily increase during the night until reaching a maximum at 07:00 UTC (of 31, 10 and 2.3 μg m$^{-3}$ respectively for nitrate, ammonium and sulfate), after which they decrease until 11:00 UTC. Between midnight and 05:00 UTC, the concomitant increase of the abundance of particle concentrations with a decrease in ammonia amounts might be associated with gas-to-particle conversion process favored by high relative humidity and low temperatures (see Fig. 8b and 8d) or eventually with the variability of particle concentrations being advected to the Paris region. In the afternoon, a second relative maximum of particle component concentrations at the surface is found around 12:00-13:00 UTC, but with lower intensity (19, 6 and 1 μg m$^{-3}$ respectively for nitrate, ammonium and sulfate). An evening peak is also remarked around 22:00 UTC for the concentrations the 3 particle species (respectively 23, 7 and 2 μg m$^{-3}$). The daily evolution after 10:00 UTC of the particle components and ammonia is rather similar, without any particular anticorrelation which does not suggest a dominant formation or volatilization of particles.

### 3.5 Vertical distribution of air pollutants over the Paris region

In this subsection, we use vertical profiles of aerosol distributions measured by a backscatter lidar at SIRTA in order to analyze the link between air pollutant concentrations at the surface, their vertical profile and total column integrated amounts. Aerosol vertical distribution is used here as a proxy for air pollution, since no measurements of the diurnal evolution of the vertical profiles of gaseous pollutants such as ammonia or specific particle components such as ammonium or nitrate are available (only possible through very specific field deployments

such as airborne in situ instrumentation or with a diode laser spectrometer on-board weather or tethered balloons with open cavity, for avoiding the problems of "sticky" nature for ammonia). We depict the average diurnal evolution over the periods P1 and P2, in terms of lidar measurements, sun photometer-derived AODs and surface $PM_{2.5}$ (see Fig.9a-d). Moreover, we extract the time series of lidar attenuated backscatter at 150 m of altitude (the lowest level at which calibrated lidar measurements are available) for depicting the hourly evolution of near

surface air pollutant content (Fig. 9e-f, blue curves) and also we use attenuated backscatter integrated (indicated as IAB - Integrated Attenuated Backscatter) over the altitude range from 150 m to 2.5 km for analyzing the corresponding variability of total column amounts (note that no aerosol layers are observed above 2.5 km, Fig. 9e-f, red curves).

During P1 (27-28 March), baseline aerosol load conditions prevail over the Paris region, with an overall low AOD

of both fine (~0.1) and coarse (~0.04) particle fractions (see fine mode AOD at SIRTA in Fig. 9a). This is also shown by lidar measurements, showing attenuated backscatter below 2.5 km of altitude (where particles are located) ranging from 2.5 $Mm^{-1}$ $sr^{-1}$ during the day up to a night maximum of 4.3 $Mm^{-1}$ $sr^{-1}$ near the surface (Fig. 9c, e). According to the evolution of both attenuated backscatter at 150 m and surface $PM_{2.5}$ concentrations, near surface aerosol loads display a relative maximum from 07:00 to 10:00 UTC (thus during the morning peak of road

traffic at the Paris megacity; Fig. 9a, e). This is followed by a progressive reduction of near-surface particle amounts from 10:00 UTC until 13:00 UTC, as the atmospheric mixing boundary layer grows from a depth of ~400 m at 10:00 UTC up to ~1400 m at 13:00 UTC (shown as a magenta dashed line Fig. 9c). A late afternoon aerosol load increase from 16:00 UTC until 19:00 UTC is also depicted by both integrated amounts (particularly backscatter) and near the surface (a small relative maximum), which corresponds to the time of the evening peak

of road traffic. The attenuated backscatter integrated from 0.15 to 2.5 km show two additional distinct peaks at 07:00 UTC and 19:30 UTC of about 5.2 $Mm^{-1}$ $sr^{-1}$, which are associated with enhancements of aerosol content from 700 m up to 1500 m of altitude (probably due to horizontal advection, Fig. 9c).

As previously remarked, P2 is characterized by a strong increase in particle load due to transboundary transport of pollution from the Benelux and west Germany (Fig. 3c, f, i). This is reflected by large enhancements of the

AOD, surface $PM_{2.5}$ and lidar backscatter (Figs. 9b, f) as compared to P1. An increase of a factor 3 of the fine mode fraction of the AOD (up to ~0.29 and ~0.32 in Paris and SIRTA, respectively) is observed, while the AOD coarse fraction remains stable (~0.04 and ~0.03 respectively in Paris and SIRTA, not shown in the figures). Integrated attenuated backscatter and surface $PM_{2.5}$ during P2 are about a factor ~2 greater than in the two previous days (P1). After a reduction from 07:00 to 11:00 UTC (also observed for ammonia total columns on 30

March, Fig.5a), integrated attenuated backscatter shows a steady hourly enhancement from 12:00 UTC until 19:00 UTC, when it displays a clear evening peak. This steady increase is also measured in terms of AOD, but in this case during all daytime (from 07:00 to 17:00 UTC). This sustained increase during the daytime for vertically integrated amounts of particles is similar to that observed for the vertically integrated amount of ammonia measured by OASIS (Fig. 7b). Meanwhile, the near surface evolution of particle content shown by both attenuated

backscatter at 150 m of altitude and surface $PM_{2.5}$ is clearly different from that of the total amount of particles (respectively blue and red curves in Fig. 9b, f). Near surface aerosol amounts depict both the morning (clearly marked for surface $PM_{2.5}$) and evening relative maxima, that may be associated with road traffic. The reduction of lidar backscatter at 150 m at 07:30 UTC is likely associated with downward mixing of cleaner air (with less aerosols) from the residual layer (Fig. 9d at 07:30 UTC above 200 m) above the mixing boundary layer. The

afternoon minor reduction of particles coincides with a slight increase in wind speed leading to vertical and horizontal mixing. An enhancement of particle amounts near the surface is only seen late in the afternoon (around 17:00-18:00 UTC). Surface ammonia concentrations show morning and evening peaks, this last one only late in the afternoon. This confirms the consistency of the differences between near surface and total amounts of particle concentrations with those between surface and total column ammonia.

The lidar profile time series reveals the link between surface and vertically integrated amounts of aerosols. It clearly depicts the typical diurnal cycle of the atmospheric boundary layer, with a growth of the mixing boundary layer from ~150 m at 06:00 UTC until ~1500 m at 14:00 UTC (see magenta dashed line in Fig. 9d, likely associated with turbulence generated by sun light surface heating). Turbulence-associated vertical dilution within the mixing boundary layer is most likely a major reason for the clear reduction of near surface concentrations of

particles between 06:00 and 14:00 UTC (and we expect the same behavior for surface $NH_3$ also mixed within the boundary layer). This reduction is not remarked for vertically integrated amounts (AOD, attenuated backscatter or total column $NH_3$) since a change in the vertical distribution does not affect the total atmospheric amount. We assume that relatively large vertically-integrated amounts of particles between 00:00 UTC and 07:00 UTC in P2 period (compared to P1) are likely linked to particle formation. This is consistent with the relative humidity

conditions (maximum of RH of 95 % above DRH of 67%, Fig. 8d) and also the low $NH_3$ total columns measured by OASIS in the early morning (07:00 UTC) during P2 period (see Fig. 7b and Fig. 5a, showing this last one early morning total columns of $NH_3$ on 28-29 March smaller than the previous days).

Moreover, it is worth mentioning that although similarly affected by vertical mixing, the evolution of integrated amounts of ammonia and particles are not necessarily linked to the same phenomena. Indeed, the sustained

daytime or afternoon enhancement of particle total atmospheric amounts (AOD and integrated attenuated backscatter) during P2 is more likely associated with the advection of larger amounts of particle pollution during the afternoon or particle formation processes, but not to volatilization (which is a sink of particles). However, the afternoon enhancement of ammonia may be reinforced by volatilization during the afternoon drier conditions (see Fig. 8d) and likely also horizontal advection of polluted air masses.

An additional analysis that highlights the major role of vertical mixing for comparing vertically integrated and surface measurements of ammonia is shown in Figure 10. For both periods, we compare the daily evolution of total column of $NH_3$ retrieved by OASIS with that of surface measurements of ammonia multiplied by the atmospheric mixing boundary layer derived from lidar measurements (magenta lines in Fig. 9c-d). This last one corresponds to the vertically integrated amount of ammonia over the mixing layer for the case of a vertically homogenous distribution of this gas. We clearly remark a very similar diurnal evolution of these two quantities in

relative terms, for both periods. This confirms the good consistency between these two independent measurements of ammonia (total column and surface data). Differences in absolute terms (between 0.5 to 1 $10^6$ molec. $cm^{-2}$) likely come from the evolution of the vertical profile of ammonia, changes with respect to the vertically homogenous distribution, and also the variability of ammonia abundance in the residual boundary layer and the

free troposphere above the mixing layer.

Finally, local/regional emission sources could be a possible explanation for the observed $NH_3$ enhancements during the afternoon. Agricultural $NH_3$ emissions are weak over the Greater Paris area, but they are strong over the adjacent Picardie and Ardenne-Champagne regions, located 30 to 150 km upwind of the Créteil measurement site for the north-easterly wind conditions during the P1 and P2 periods (Fortems-Cheiney et al., 2020, Fig. 4c

based on detailed emission modelling). The diurnal $NH_3$ emission variation is strongly temperature dependent as shown among others by Hamaoui-Laguel et al., (2012) from simulations with a mechanistic emission model (Volt'Air, Génermont et al., 2018). Advection of these emissions to the Créteil site could be rapid enough to explain the observed afternoon $NH_3$ column increase, given the surface wind speed of 3 to 5 m/s and probably larger winds at altitude.


## 4.  Conclusions

We have carried out a comprehensive analysis of the diurnal evolution of ammonia amounts at the surface and over the total atmospheric column during a springtime pollution outbreak at the Paris megacity, considering different factors and variables that influence their variability at surface level and in altitude. Using remote sensing,

meteorological and chemistry-transport models, we have described the regional atmospheric conditions over western Europe affecting air quality over the Paris region during late spring 2012. A clear picture of particulate pollution within the Paris region was drawn from in situ surface measurements of PM from the Paris region operational network. These results allowed us to distinctly identify two phases within the pollution outbreak in Paris: local formation of rather moderate pollution on 26-27 March 2012 (P1) and the arrival of relatively large

amounts of transboundary pollution from the Benelux and west Germany on 28-30 March 2012 (P2), leading to high surface $PM_{2.5}$ concentrations (up to 80 μg m$^{-3}$).

The daily evolution of ammonia in the Paris megacity was characterized by state-of-the-art measurements from the AiRRmonia surface in situ instrumentation and remote sensing of total atmospheric columns from the OASIS observatory. To the authors' knowledge, this is the first study analysing the daily evolution of ammonia total

columns with high temporal resolution (10 minutes in cloud free conditions) over a megacity. Clearly different evolutions of ammonia concentrations at the surface and integrated over the atmospheric column were observed. Ammonia total columns during the late March 2012 pollution event depicted a clearly steady diurnal enhancement on each of the days of the event, during most of daytime (2 days) or the afternoon (3 days). On the other hand, surface ammonia measurements during this event principally revealed rather moderate fluctuations with

significant morning time peaks.

Despite a wide variety of factors influencing ammonia, our study distinctly identifies a crucial role of vertical mixing within the atmospheric boundary layer for explaining the difference between the evolution of ammonia at the surface and that integrated over the total column. Indeed, the growth of the mixing boundary layer from 150 m deep at 06:00 UTC up to 1500 m deep at 14:00 UTC entrains vertical dilution of atmospheric pollutants within

the boundary layer and thus a relative reduction of air pollutants concentrations near the surface (but not over the total atmospheric column). By comparing surface (or near surface) and total column amounts, we observe a similar behaviour for both ammonia and particles. Both for P1 and P2, surface concentrations for these two pollutants mainly depict only morning and late afternoon peaks, while total columns show a steady enhancement particularly in the afternoon. Vertical dilution is then likely responsible for a prevailing reduction of surface concentrations

until 14:00 UTC (explaining that they do not depict enhancements). Other processes such as surface and canopy uptake from surrounding ecosystems, depending on pH, temperature, light and total nitrogen input, may also explain surface concentration reductions (Massad et al., 2010, Flechard et al., 2013, Personne et al., 2015). Afternoon enhancement for surface amounts is only seen later in the afternoon (16:00-17:00 UTC). Moreover, the

joint analysis of the evolution of ammonia, ammonium and nitrate highlighted the occurrence of volatilization of these last two to release ammonia in the atmosphere during the afternoon of P1. When it comes to P2, the evolution of total column amounts of ammonia and particles in the Paris region seems to be mainly driven by the arrival of polluted air masses originating from the Benelux. Low relative humidity (clearly below the deliquescence point of ammonium nitrate) during the afternoons of the last period also suggests the possible volatilization for enhancing ammonia concentration (although this is not clearly seen as a major driver of measured nitrate/ammonium concentrations). However, the diurnal variation of $NH_3$ emissions could be one of the factors leading to afternoon maxima, as emission sources are strong in the vicinity and upwind (under given conditions) of the Paris region. Night-time ammonia and ammonium during P2 indicates gas-to-particle formation, which could also occur at higher altitudes (due to higher relative humidity, not shown), leading to distinct lower total column values of ammonia in the early morning (for 2 days). This issue would be best addressed with chemistry-transport model simulations and dedicated in situ measurements (including nitric acid, particulate nitrate and ammonia) for the parametrization and validation of the model.

Our comprehensive study illustrates the benefit of using together total column and surface measurements of ammonia for understanding how vertical mixing within the atmospheric boundary layer influence the daily evolution of ammonia. This work also confirms the role of temperature and relative humidity for ammonia volatilization and particle formation.

For a particular geographical location, ground-based instruments in urban sites such as OASIS with high temporal resolution provide highly valuable information on the diurnal evolution of atmospheric species (especially gaseous pollutants). Ground-based remote sensing is also very valuable for validating satellite retrievals since both typically derive total column amounts of atmospheric species, which may significantly differ from their abundance at the surface.

The results of this study highlight the need of a better chemical characterization for comprehensive understanding of gas-particle partitioning over the column. A quantitative ammonia-ammonium equilibrium throughout the atmospheric column (as function of altitude) should be considered from dedicated in situ measurements field campaigns. This may be addressed, for instance, by the development of spectroscopic instrumentation onboard standard weather or tethered balloons, capable of simultaneously measuring the vertical distribution of ammonia and particles components, in combination with chemistry transport models, as already developed for greenhouse gases (such as $CO_2$, $CH_4$ and $H_2O$, see Joly et al., 2020).

**Author contribution**

RDK is the main author of the paper, wrote the text, made most of the figures and analyzed the data. JC, PC, JEP and MB contributed to the manuscript writing, discussions and analysis of the figures. JC made one of the figures of the paper. JEP provided data and carried out the ISORROPIA II calculations. PC, MR and XL operated the OASIS observatory. BT made an initial analysis of OASIS data. JCD provided observational data and AR the ESMERALDA/CHIMERE outputs. FH and JO provided support on the anlaysis and the PROFFIT code for processing OASIS dataset and deriving NH3 data.

**Competing interests**

The authors declare that they have no conflict of interest.

**Acknowledgments**

The authors from LISA acknowledge support from CNES (Centre National des Etudes Spatiales) and the
INSU/CNRS (Institut National des Sciences de l'Univers/Centre National de Recherche Scientifique) in the
framework of the projects IASI-TOSCA (Terre Ocean Surface Continental Atmosphère) and LEFE-CHAT as
well as the OSU-EFLUVE (Observatoire des Sciences de l'Univers-Enveloppes Fluides de la Ville à
l'Exobiologie) and the University Paris Est Créteil for the routine operation of the OASIS observatory. The
research was also funded by DIM Qi2 (Paris Region). A particular acknowledgement shall be given to the
collaborator of the KIT in Karlsruhe, Germany, for their continues support and involvement. Work at IMK has
been funded by the ATMO program of the Helmholtz Association of Germany Research Centres. The authors
wish to thank AirParif and SIRTA for in-situ data and ground-based lidar measurements, and the NASA Goddard
Space Flight Center for providing the temperature and pressure profiles of the National Centers for Environmental
Prediction (NCEP) for the OASIS retrievals of $NH_3$. Furthermore, our thanks extend to AIRPARIF and their
provision of the ESMERALDA output based on CHIMERE, used in this analysis, as well as the IPSL, providing
the ERA-Interim reanalysis data that are accessible via the CLIMSERV platform by download. Thanks is extended
to the NASA Terra MODIS for providing data on their platform as well as the French data government website,
which provided shape files of the land use and were visualized with the Q-GIS software. Finally, the authors want
to acknowledge and thank AERONET for the provision of the sun photometer data.

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

**Table 1.** Correlation of different available PM values of Vitry, Gennevilliers, Bobigny and SIRTA sites during 26-27 March (period 1) and 28-30 March (period 2). SE refers to Standard Error, and $R^2$ to the square of the correlation coefficient.

| | 26-27 March (Period 1) | | | 28-30 March (Period 2) | | |
|---|---|---|---|---|---|---|
| | Slope | SE | $R^2$ | Slope | SE | $R^2$ |
| $PM_{2.5}$ Vitry vs $PM_{2.5}$ Gennevilliers | 0.84 | 0.09 | 0.67 | 0.91 | 0.04 | 0.86 |
| $PM_{2.5}$ Vitry vs $PM_{2.5}$ Bobigny | 0.89 | 0.10 | 0.63 | 0.94 | 0.04 | 0.91 |
| | | | | | | |
| $PM_{10}$ Vitry vs $PM_{10}$ Gennevilliers | 1.04 | 0.09 | 0.74 | 1.04 | 0.05 | 0.84 |
| $PM_{10}$ Vitry vs $PM_{10}$ Bobigny | 1.02 | 0.11 | 0.63 | 0.93 | 0.07 | 0.73 |
| | | | | | | |
| $PM_1$ SIRTA vs $PM_{2.5}$ Vitry | 0.57 | 0.06 | 0.67 | 0.49 | 0.08 | 0.35 |
| $PM_1$ SIRTA vs $PM_{2.5}$ Gennevilliers | 0.61 | 0.04 | 0.82 | 0.44 | 0.09 | 0.26 |
| $PM_1$ SIRTA vs $PM_{2.5}$ Bobigny | 0.57 | 0.04 | 0.84 | 0.48 | 0.08 | 0.34 |

**Table 2.** Meteorological variables at the surface and AOD on 26-27 March (period 1) and on 28-30 March (period 2), displayed by the median, minimum and maximum values, for temperature, relative humidity, deliquescence relative humidity, wind speed and –direction as well as AOD in its fine mode (FM) and coarse mode (CM).

|  | Period 1 | | | Period 2 | | |
|---|---|---|---|---|---|---|
|  | Median | Minimum | Maximum | Median | Minimum | Maximum |
| Temperature (°C) | 16.2 | 10.8 | 21.8 | 14.8 | 10.5 | 21.0 |
| Relative Humidity (%) | 41.8 | 19.0 | 66.0 | 54.8 | 26.5 | 77.0 |
| Deliquescence Relative Humidity (%) | 70.9 | 65.1 | 76.7 | 72.1 | 65.6 | 77.2 |
| Wind speed (m/s) | 3.34 | 0.86 | 8.19 | 2.51 | 0.29 | 7.09 |
| FM AOD 550nm – Paris | 0.10 | 0.05 | 0.14 | 0.29 | 0.13 | 0.55 |
| CM AOD 550nm – Paris | 0.04 | 0.02 | 0.07 | 0.04 | 0.02 | 0.13 |
| FM AOD 550nm – SIRTA | 0.10 | 0.04 | 0.15 | 0.32 | 0.10 | 0.62 |
| CM AOD 550nm – SIRTA | 0.04 | 0.01 | 0.08 | 0.03 | 0.02 | 0.17 |


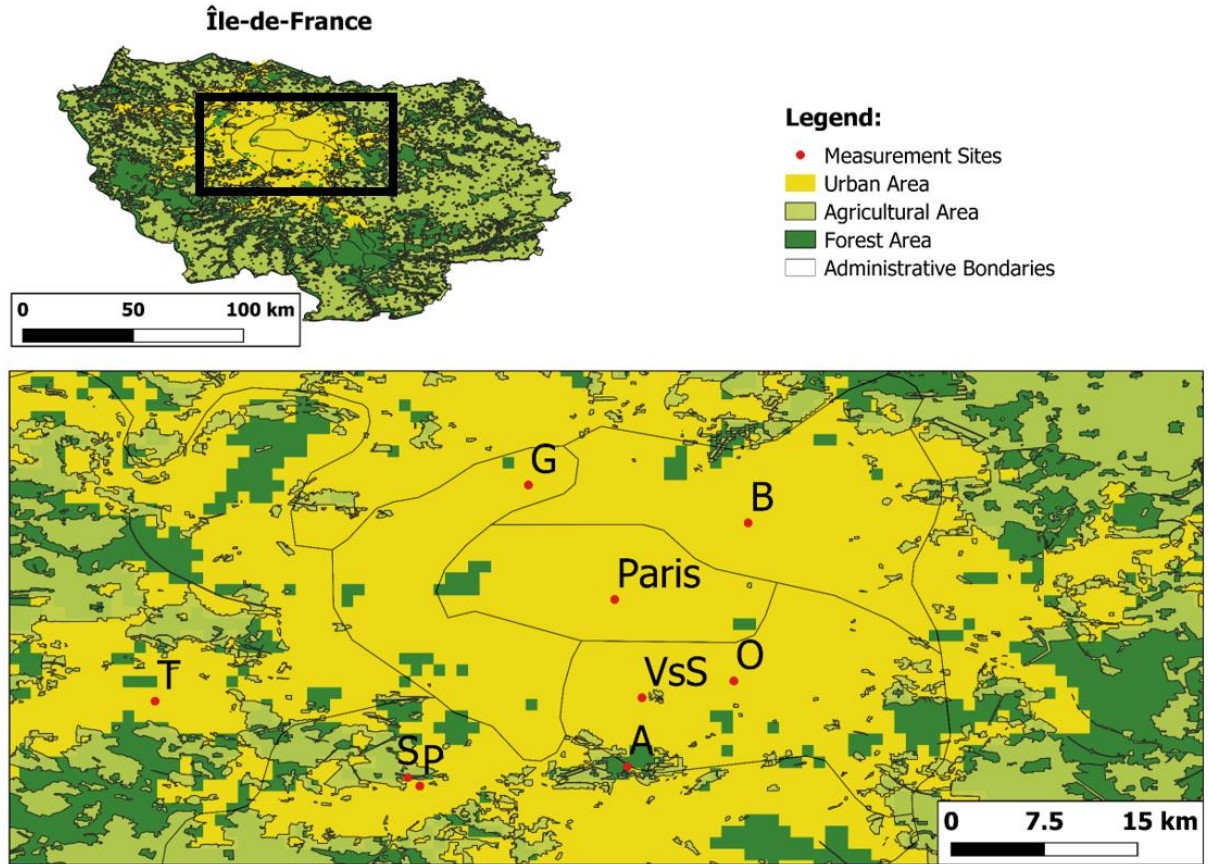

**Figure 1.** Outline of Paris Region and a zoom into relevant sites (A – Airport Orly, B – Bobigny, G – Gennevilliers, O – OASIS, P – Palaiseau, S – SIRTA, T – Trappes, VsS - Vitry-sur-Seine) using shape files provided by data.gouv.fr (https://www.data.gouv.fr/fr/datasets/espaces-agricoles-de-la-region-ile-de-france-inscrits-sur-la-cdgt-du-sdrif-arrete-en-2012-idf/#discussion-5cc30bdb8b4c4166219c058e, last access: 30 April 2019) and processed with QGIS 3.6.

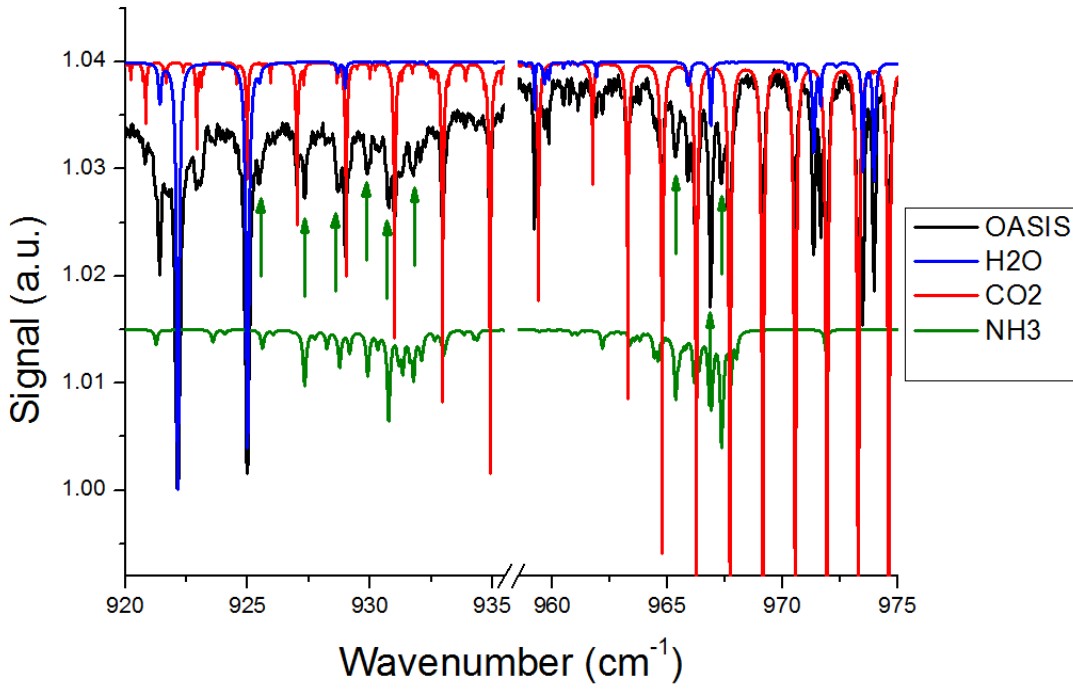


**Figure 2.** OASIS-FTIR atmospheric spectrum recorded with the BRUKER Vertex 80 at Creteil on 21 March 2012 in the two microwindows (before and after the spectral gap), showing the strong ammonia absorbing lines (pointed out by the green arrows) around 932 $cm^{-1}$ (origin of the NH3 $\nu$2 band) with individual contributions of the main interfering species represented from the atlas of Meier et al. (2004). All spectra are plotted using arbitrary unit

(a.u.) on the Y scale.

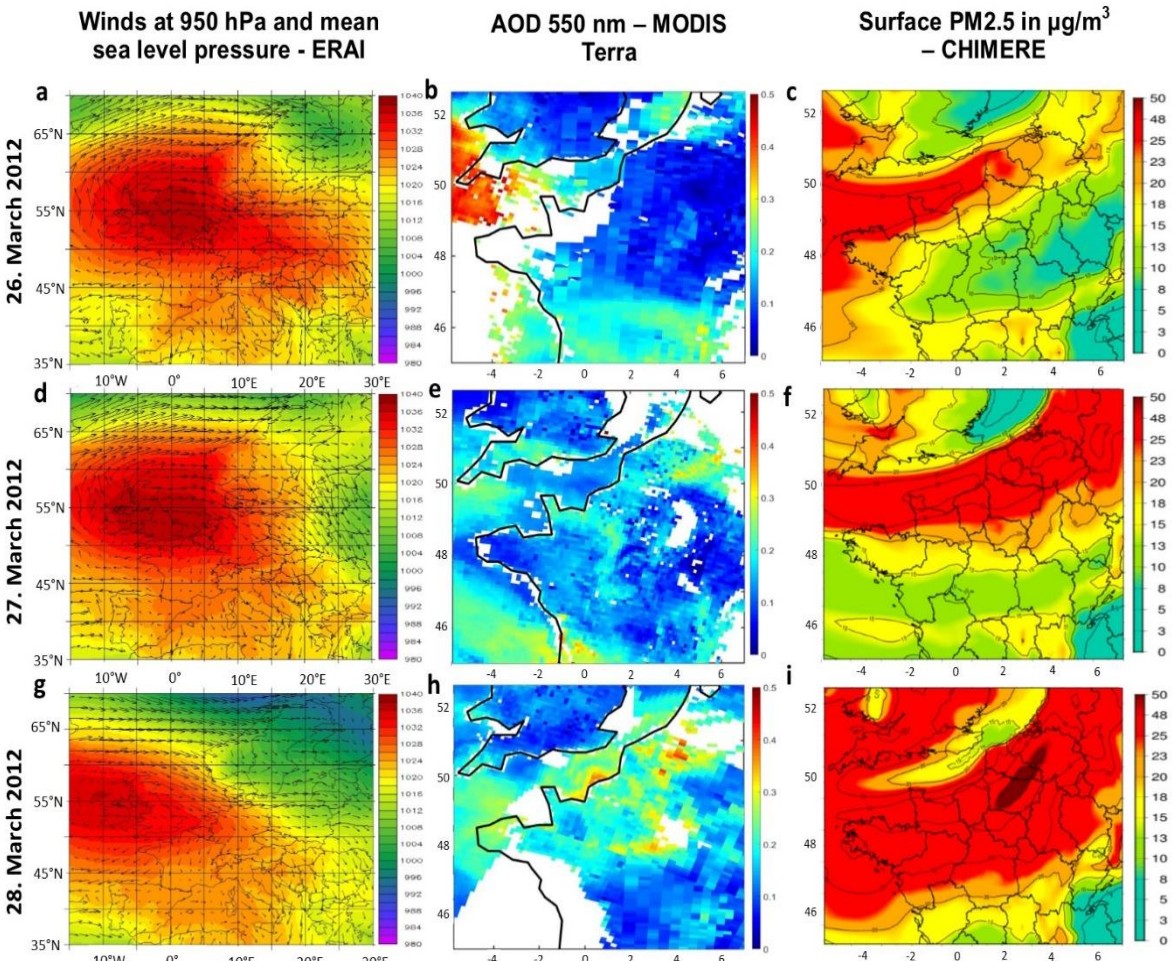

Figure 3. **(a, d, g)** Meteorological conditions characterized by 950 hPa winds (arrows) and sea level pressure (shading) over Europe (15° W to 20°E and 40° to 65°N) from ERAI reanalysis in 26-28 March 2012. Horizontal distribution of particles over northern France (-5° to 7°E and 45° to 52.5°N) in terms of **(b, e, h)** AOD at 550 nm from MODIS on-board the Terra satellite and **(c, f, i)** surface $PM_{2.5}$ in µg/m$^3$ from the CHIMERE model in the period 26-28 March 2012.

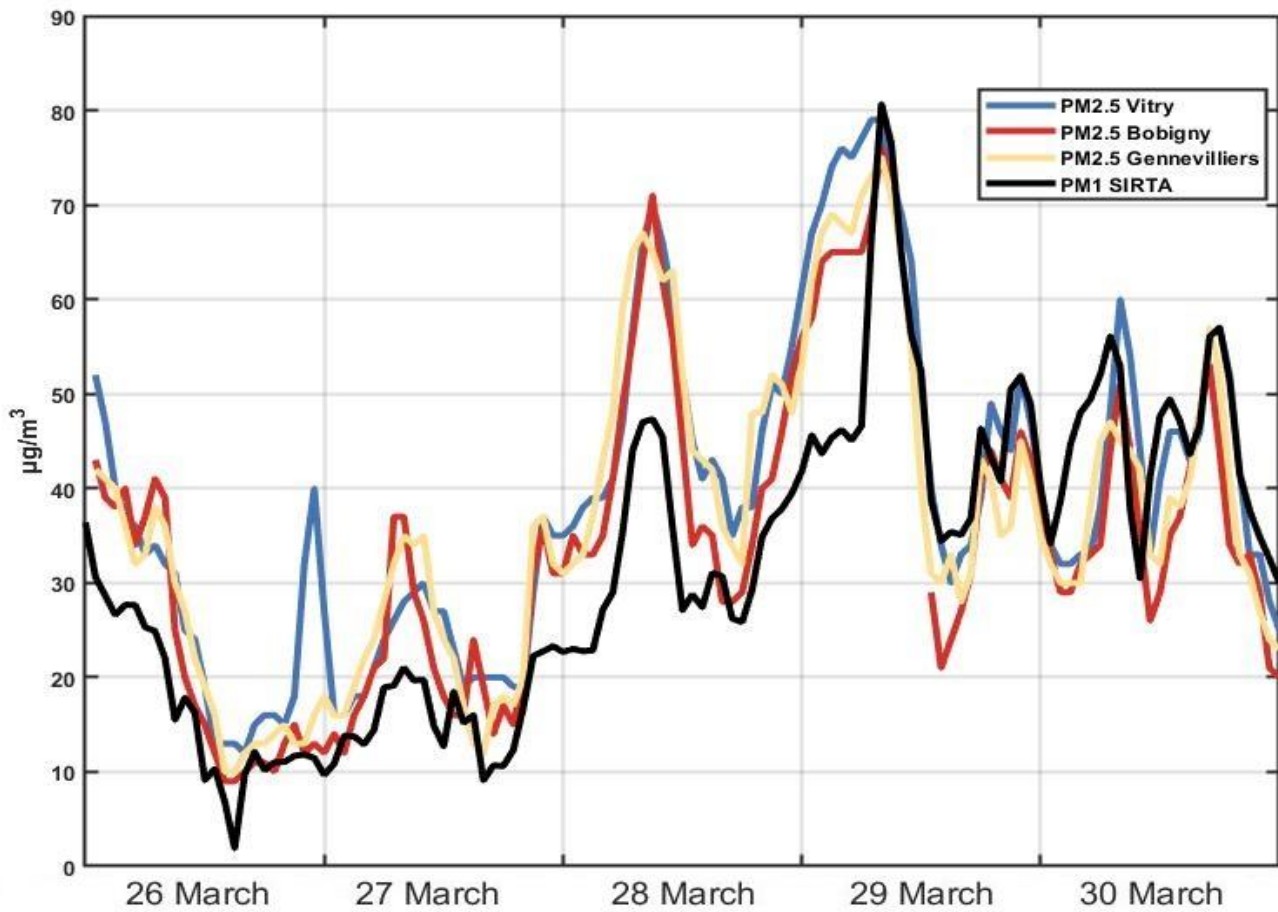

**Figure 4.** Particle matter concentrations measured at the surface at the station of Vitry, Bobiny, Gennevilliers and SIRTA, respectively southeast, northeast, northwest and southwest of Paris. Particle concentrations in terms of $PM_{2.5}$ and $PM_1$ are provided respectively at the Airparif three stations and at the last one in the period between 26-31 March 2012.

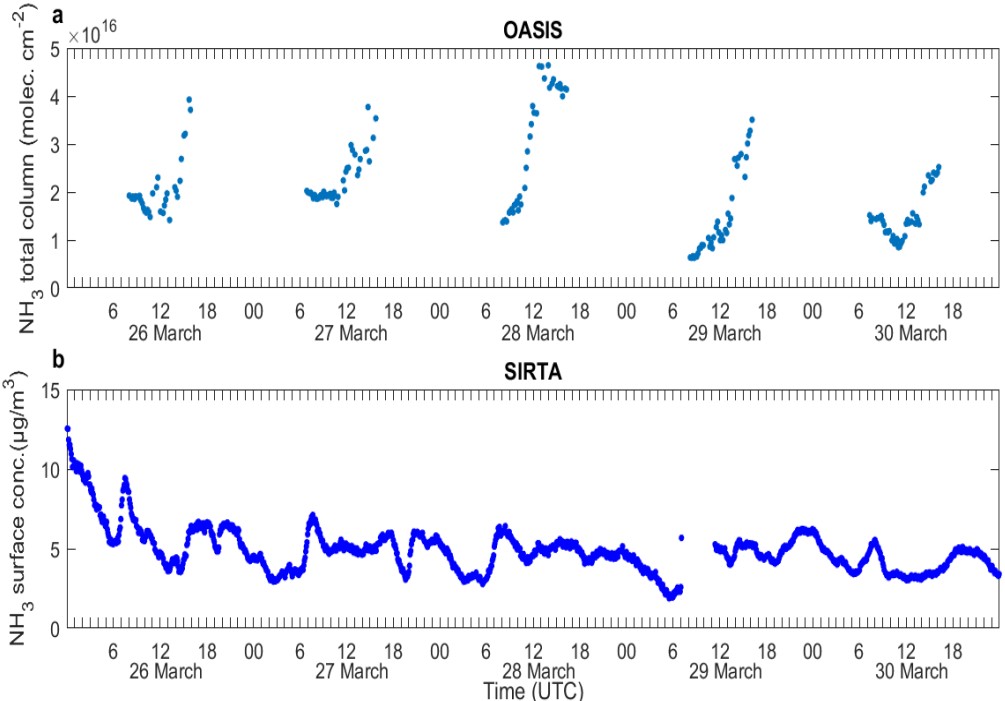

**Figure 5.** Observations of atmospheric ammoniac concentrations over the Paris region from 26 to 30 March 2012. The upper panel **(a)** displays total column retrievals at Créteil derived from OASIS observatory (48.79N 2.44E) measurements during the day (~07:00 and 16:00 UTC). The lower panel **(b)** displays continued ammonia surface concentration measurements from the AiRRmonia instrument near Palaiseau (SIRTA observatory, 48.71N 2.20E). This figure shows all available individual measurements from OASIS FTIR instrument and AiRRmonia in situ analyser.

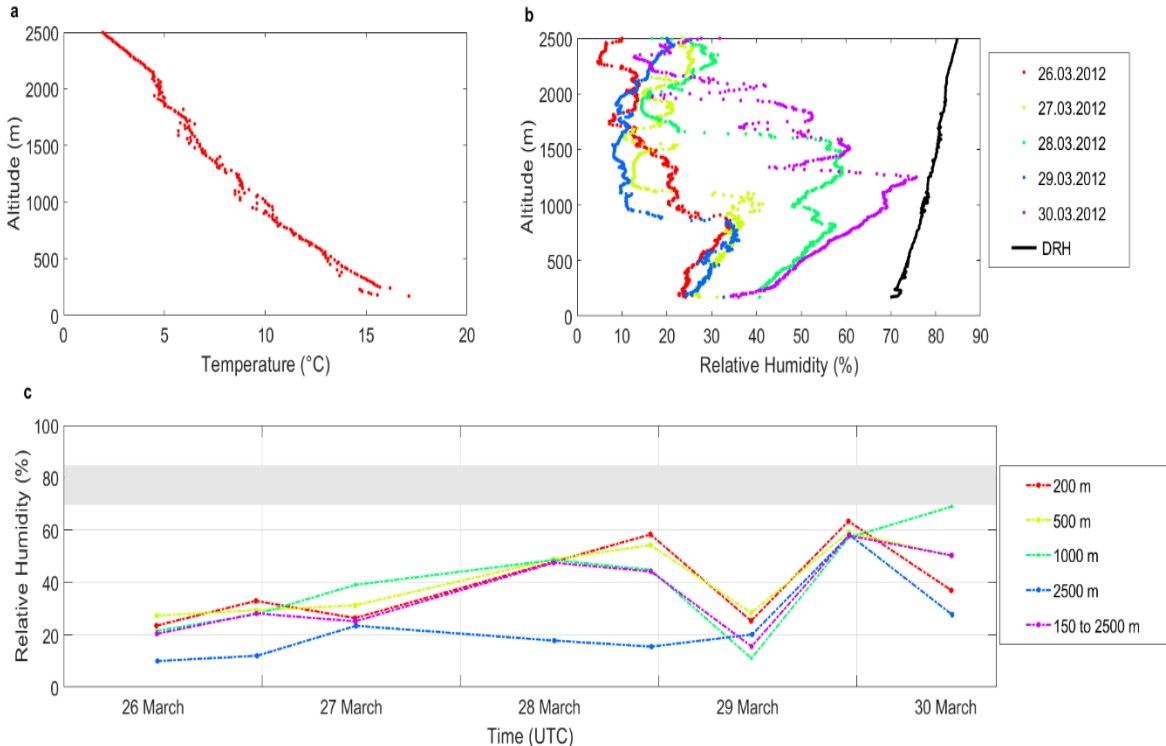

**Figure 6. (a)** Median temperature (°C) over the period of interest in the vertical during mid-day. **(b)** Relative humidity (%) in the vertical (mid-day only) from Trappes station and DRH based on the median temperature from (a). **(c)** Relative humidity evolution at different heights (average over ±25 m) for all radio-soundings on 26-30 March, whereby the grey bar indicates the DRH lowest and highest values corresponding to 170 m and 2500 m altitude. DRH refers to Deliquescence Relative Humidity.


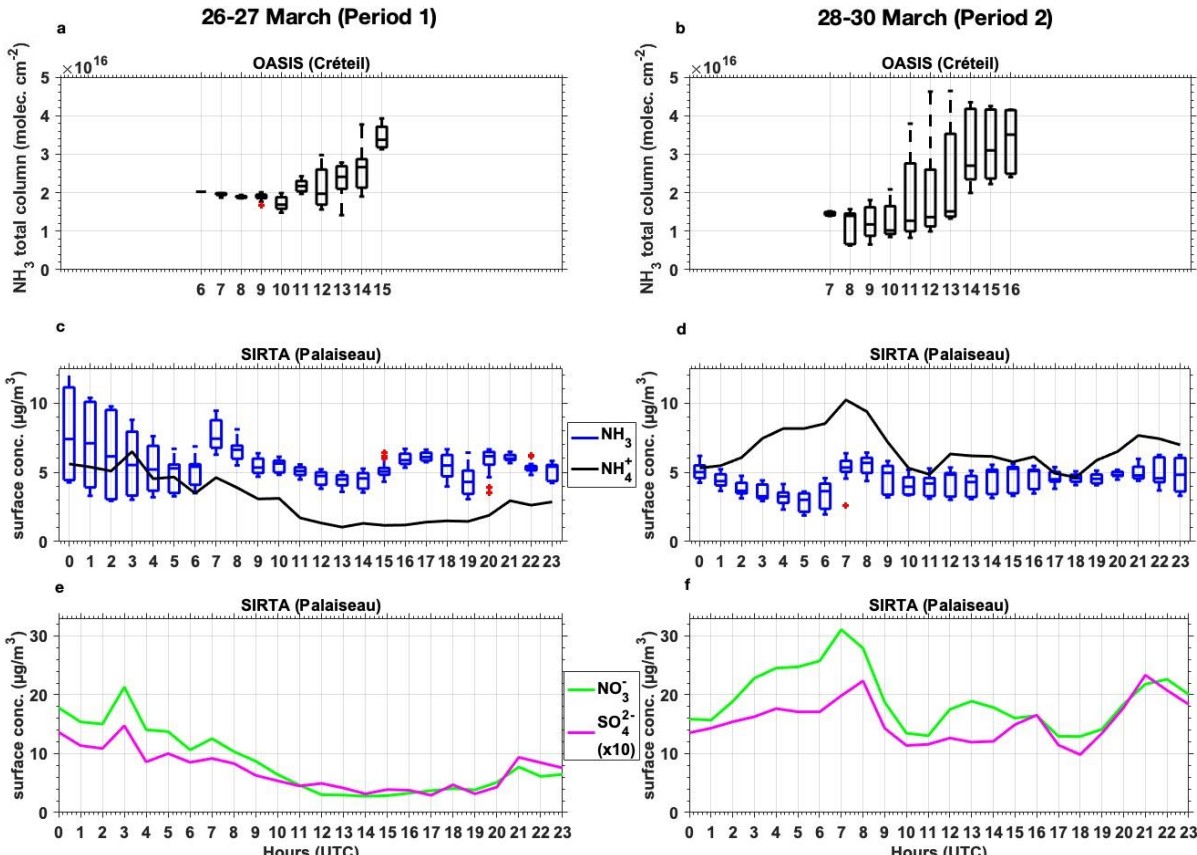

**Figure 7**. Average diurnal evolution of total column NH$_3$ **(a, b)**, surface NH$_3$ and NH$_4^+$ **(c, d)**, hourly median surface measurements of NO$_3^-$ and SO$_4^{-2}$ **(e, f)** for period 1 on the left side and period 2 on the right side. Hourly boxplots of NH$_3$ total column retrieved from OASIS (a, b) and hourly boxplots of NH$_3$ from surface measurements show within the boxplot the median as a line in the plot, 25th and 75th percentile as the lower and upper border of the box and whiskers extended to most extreme data points, whereby outliers are separately marked with a '+'.

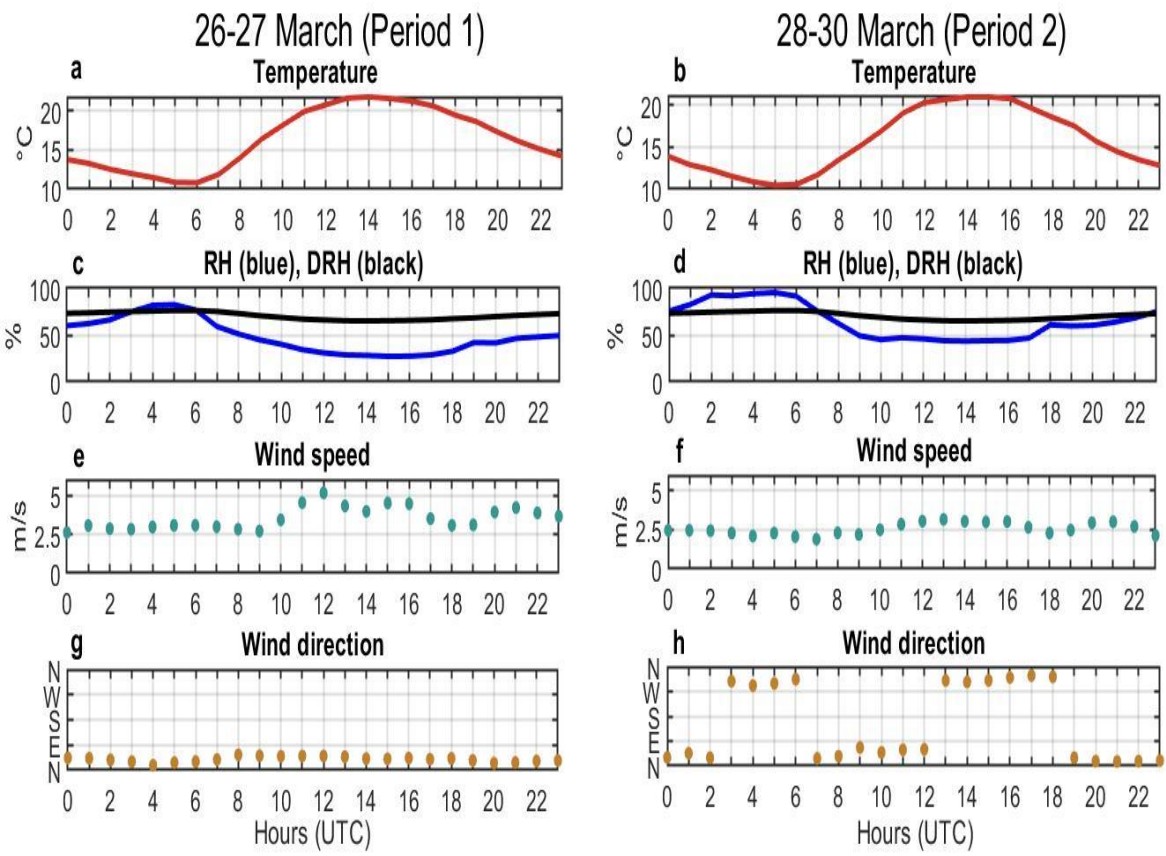


**Figure 8.** Hourly median surface temperature in °C **(a, b)**, relative humidity (blue) and calculated deliquescence relative humidity (DRH; black) in % **(c, d)**, wind speed in m/s **(e, f)** and wind direction in degrees **(g, h)** are presented in the left column for period 1 and in the right column for period 2. Whereby temperature and relative humidity measurements origin from OASIS, while wind speed and wind direction origin from SIRTA supersite.


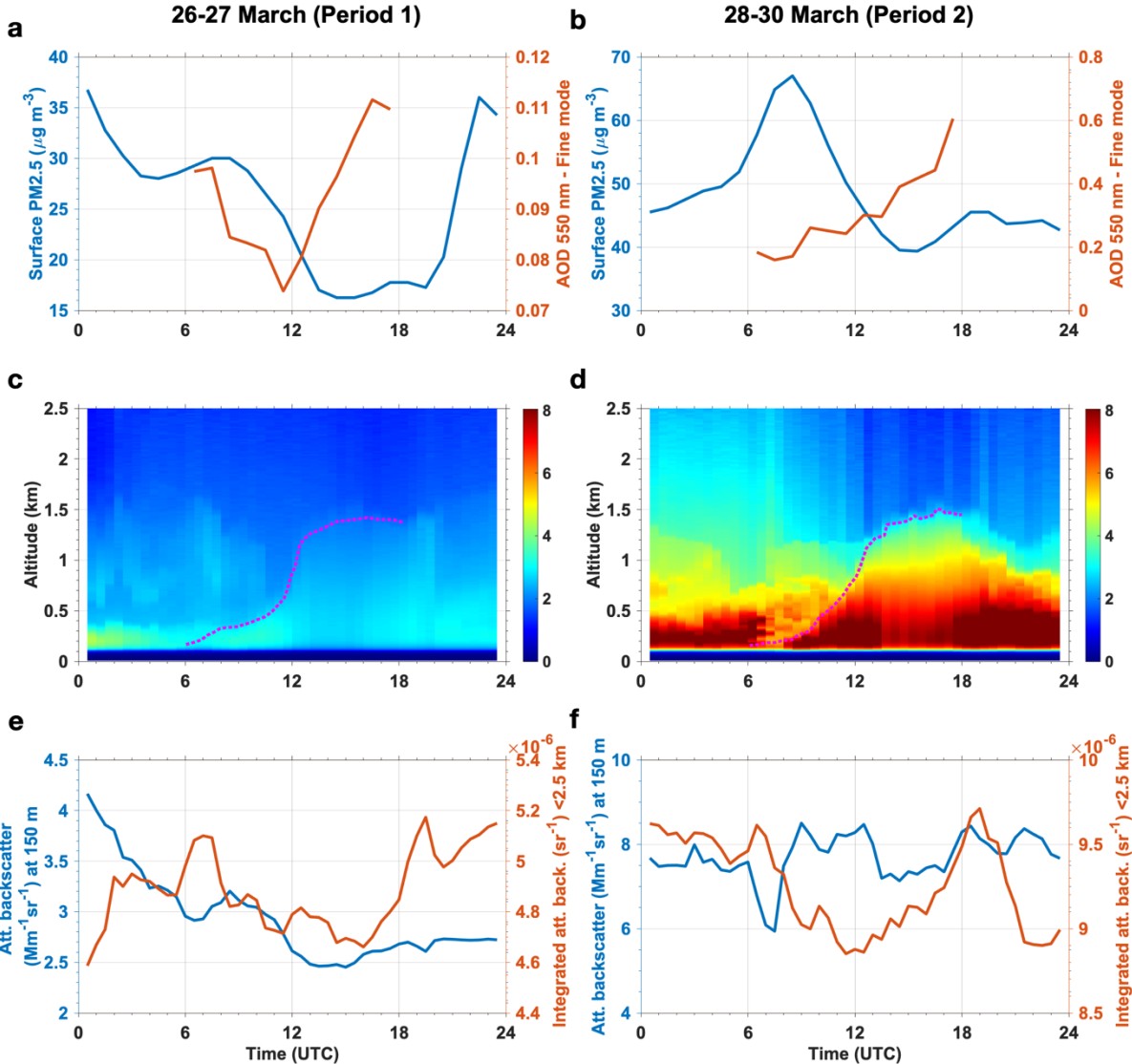

**Figure 9.** Median diurnal evolution of the AOD of the fine mode fraction of aerosols at 550 nm at SIRTA (red
lines) and surface measurements of PM2.5 at Vitry (blue lines) during the periods **(a)** 26-27 March 2012 (P1) and
**(b)** 28-30 March 2012 (P2). Median diurnal evolution of vertical profiles of attenuated backscatter measurements
at 355 nm of a ground-based lidar at SIRTA for depicting the vertical distribution of aerosols during **(c)** P1 and
**(d)** P2. Dashed magenta lines in (c) and (d) show the top of the mixing boundary layer manually tracked as the
lowest discontinuity of the lidar profiles during daytime (06:00 – 18:00 UTC). Lidar-derived proxies of the diurnal
evolution of particles over the total column (attenuated backscatter integrated between 0.15 and 2.5 km of altitude,
in red) and near the surface (attenuated backscatter at 0.15 km, in blue) for **(e)** P1 and **(f)** P2. For clarity,
measurements in panels (a, b) are shown with hourly time resolution whereas it is of 30 minutes for panels (c-f).

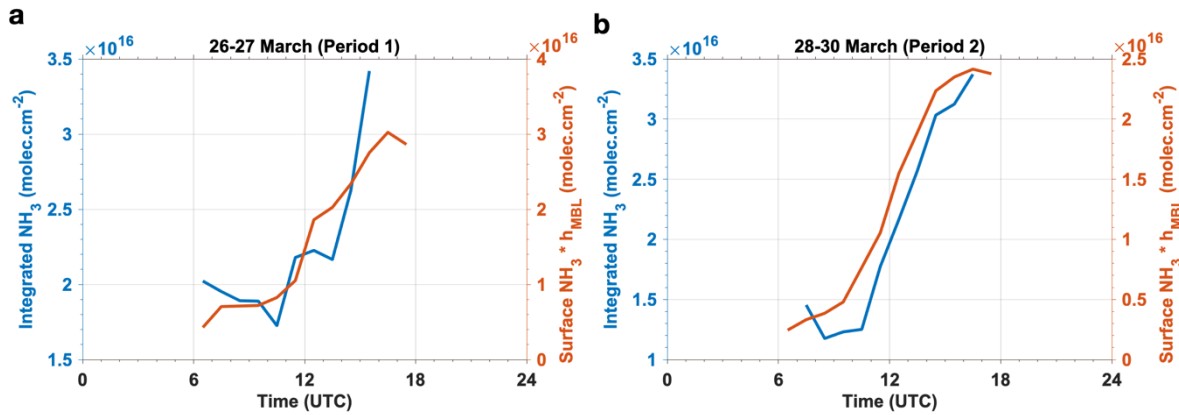


**Figure 10.** Diurnal evolutions of total column integrated ammonia concentrations retrieved by OASIS (blue lines) and surface concentrations measured in situ at SIRTA multiplied by the mixing boundary layer derived from lidar measurements (red lines) averaged over the periods **(a)** 26-27 March 2012, P1, and **(b)** 28-30 March 2012, P2. This last amount provides an estimate of the vertically integrated ammonia abundance over the atmospheric mixing boundary layer heights (those shown in

Fig. 9c-d) in the case of a vertically homogenous distribution of this gas.