# Peer review of "Diurnal evolution of total column and surface atmospheric ammonia in the megacity of Paris, France, during an intense springtime pollution episode"

_Atmospheric Chemistry and Physics, 2020_

## Referee Comment (RC1) · Anonymous Referee #2 · 18 Feb 2021

This paper investigates ammonia diurnal variability near surface as well as in the tropospheric column over the megacity of Paris in spring 2012 during an intensive pollution episode and shows significant differences between them. The observations are analyzed in conjunction with particulate matter levels and meteorological parameters and the differences are explained by the dilution within the boundary layer and also by volatilization of ammonium nitrate particles. The study is interesting and contributes to the understanding of ammonia variability in the atmosphere. The manuscript is overall nicely written; however, some clarifications and corrections are needed before publication in ACP.

Line 382-384: This sentence is misleading. 'higher altitudes' needs definition because Fig 6. shows that this anti-correlation between RH and Temp is limited to the first 1000 or 1500 m.

Line 235: Please provide further information on the model version you are using is it running off-line and if yes with what meteorological parameters. Are the CHIMERE simulations associated with ERA-Interim meteorology mentioned in lines 238-342 to be used to analyze the meteorological conditions?

Line 237: The reference you provide is an entire textbook. Please be specific. Which thermodynamic model is used in that simulation. (Unfortunately, the web site provided for the model in line 234 requires password, so it seems to be useless for the reader. I suggest removing it.)

Line 372: ISORROPIA reference is Nenes et al. 1998; ISORROPIA II reference should be Fountoukis and Nenes, ACP, 2007 please correct accordingly. Also, in which form ISORROPIA was run (forward) or backward? What input data have been used? Information on how the ISORROPIA (or ISORROPIA II) simulations are done is missing and will affect any results of the model, although in the paper such results are discussed only qualitatively.

Line 354-355: figure 6a shows temperature profile up to 2.5km, the reference to the tropopause level is misleading, rephrasing is needed.

Line 285: the discussed PM2.5 levels are the results of simulations or observations ? do in-situ observations show similar levels?

Line 577-580: Does your model reproduce such behavior during night? The model results need to be discussed more and valorized. Does the model reproduces observed surface and profiles of NH3? The reader remains with the question why not comparing the model results to the observations?

Line 241-242: Provide temporal resolution. Are these data used as input to Chimere model?

Line 132: particulate matter

Line 211: the following

Line 297: homogeneous

Line 375: decreasing ammonia

Lines 507-508: 'depicts... depicting.. the last one...' (which one?), please rephrase the sentence.

---

## Short Comment (SC1) · 9 Apr 2021

**Benjamin Loubet**

benjamin.loubet@inrae.fr

Received and published: 9 April 2021

General comments This manuscript presents ammonia total column concentration measured above Paris with an FTIR instrument (OASIS) during an intense springtime pollution episode in March 2012. The main object of the manuscript is to discuss the diel cycle of the total column NH3 concentration observed, which shows a flat profile in the morning and a steady increase in the afternoon, contrary to surface NH3 concentrations (measured at around 30 km south), that show a rather flat cycle. The

manuscript provides a large overview analysis making full use of LIDAR, and surface aerosol concentration and composition, measurements as well as meteorological informations, to try explaining the total NH3 column diel cycle. The manuscript shows clearly that the surface NH3 and particulate matter concentrations are decoupled from the total column concentrations, which is a clear and important outcome of this work.

The manuscript is well structured and written and make appropriate reference to the literature (some more references are given in the detailed comments). The data presented are of great interest for the community to better understand the relationship between surface concentration and satelitte measurements for ammonia, which is a key pollutant in developped countries.

The main conclusion raised by the authors is that the observed total NH3 column diel cycle may be due to a combination of dilution (linked to the atmospheric boundary layer height cycle), aerosol volatilisation, and ammonia emissions at the regional scale. However, no clear quantitative evidence is shown to strengthen either of these conclusions, but rather qualitative arguments are given. This is the main weakness of the manuscript, which to my view, should be addressed prior to publication.

In particular, I would suggest better quantifying the effect of relative humidity and temperature on the ammonia-to-ammonium-nitrate equilibrium, and showing a graph of these dependencies. I also suggest trying to calculate, if possible, the averaged column concentration of ammonia and particles using the boundary layer height retrieved from the LIDAR measurements. I am not sure this is fully feasible, but even with some uncertainties; this may help comparing the column averaged "concentration" with the surface concentration. Additionally, the authors should bear in mind that the surface concentration of ammonia might be very much influenced by the surface surrounding the measurements, which in this case is probably a mix of forests and crops, and hence would lead to an active sink.

Detailed comments Line 35: vertical dilution is not per se a variable but a process. The
boundary layer height may be the variable to add here. Lines 58-59: Other effects of particle transport and deposition are acidification and loss of biodiversity (e.g. Sutton et al. 2011). Lin 59-61: This sentence on volatilisation of ammonium nitrate particles may benefit from being moved at the end of this paragraph or maybe even later in the text. Line 65: please provide the year of the evaluation of the total ammonia emission in Europe. Line 67: ammonia volatilization also depends on soil conditions like for instance humidity and pH. Line 71-72: Fires are also probably a significant source of ammonia in South East Asia. Please give a range of significance of this source worldwide. Line 82-83: Although it is true that agricultural activities are one of the key factor, explaining the rise in particle pollution it may be important to stress that other precursors are also essential in the process, like for instance nitrogen oxides emissions. Line 91-94: please consider reading this paper regarding stickiness of ammonia and inlet materials: Whitehead, J. D., Twigg, M., Famulari, D., Nemitz, E., Sutton, M. A., Gallagher, M. W., and Fowler, D.: Evaluation of laser absorption spectroscopic techniques for eddy covariance flux measurements of ammonia, Environmental Science & Technology, 42, 2041-2046, 2008. Line 169: please explain what you mean by "co-add 30 scans". Line 176-177: it would be important here better explaining how the nitric acid HNO3, in particular, is taken into account since these compound made directly interact with ammonia in the atmosphere. Although it is explained in Trounadre et al. (2020), it would be also important here to quantify the water vapour interference here. Line 182: Retrieval error that is given between 20 and 35% may be dependent on one or two compounds mostly. It would be good to explain the weight of water vapour and of the vertical profile shapes on this error. Line 192: It may be worth precising here that ammonia is first absorbed in acid solution. Line 208: The word "pre-conditions" is unclear. Could you please rephrase? Line 205-219: Since equilibrium between ammonia and ammonium aerosol is key in this manuscript I would suggest precising quantitatively this process. Indeed, in the current manuscript, it is said that when RH is much lower than DRH, particulate volatilization is favoured. However, volatilisation depends on the difference between RH and DRH I guess. I would therefore suggest adding a graph
showing how the equilibrium between ammonia and ammonium nitrate may be dependent on relative humidity. A representation like what is shown by (for instance) Nemitz (Figure 8 of the following paper) would be interesting. Alternatively, the ISORROPIA box model could be used for that. Nemitz, E., Sutton, M. A., Wyers, G. P., Otjes, R. P., Schjoerring, J. K., Gallagher, M. W., Parrington, J., Fowler, D., and Choularton, T. W.: Surface/atmosphere exchange and chemical interaction of gases and aerosols over oilseed rape, Agric. For. Meteorol., 105, 427-445, Doi 10.1016/S0168-1923(00)00207-0, 2000. Line 220: I would have expected the opposite dependence of evaporation of ammonium nitrate to pH, since water solution a lower pH leads to lower hip operation. However, this may not be true in an aerosol, which is not my domain of expertise. Line 236: Please consider replacing "exist" by "are simulated". Line 263: Please consider replacing "manually" by "visually". Line 300: this is very interesting to see that PM1 is 30% lower and PM 2.5 during this episode would it be possible to propose some hypothesis ? Line 308: Please consider replacing "denominate these two regimes as ..." by "name these two regimes ..." Line 320 and Figure 4: It would be worth showing the ratio of PM1 to PM2.5 in the Parisian area. Line 355: it is unclear if the Oasis inversion methodology uses this water vapour profile. Would this be of any help to better estimate water vapour interference? Line 372-374: it is a shame that these ISORROPIA model outputs are not shown. As already proposed in an earlier comment it would be worth showing the effect of temperature and relative humidity on the equilibrium between ammonia and ammonium nitrate. Line 377: the meaning of the sentence "vertically decreasing variation of ammonia with increasing altitudes" is unclear. Please rephrase. Line 420: It may be worth looking at the following paper that reported ammonia emissions and concentrations during the very same period in the west of Paris: Personne, E., Tardy, F., Genermont, S., Decug, C., Gueudet, J. C., MASCHER, N., Durand, B., Masson, S., Lauransot, M., Flechard, C., Burkhardt, J., and Loubet, B.: Investigating sources and sinks for ammonia exchanges between the atmosphere and a wheat canopy following slurry application with trailing hose, Agric. For. Meteorol., 207, 11-23, 10.1016/j.agrformet.2015.03.002, 2015. Lines 422-249:
The manuscript would benefit a lot if some more quantitative estimation of the effects of the boundary layer height on the concentration of the total ammonia column concentration could be estimated even roughly. Would it be feasible to use the atmospheric boundary layer height as retrieved from the LIDAR measurements shown in figure 9 to estimate an averaged ammonia concentration in the column? Lines 440-445: it should be stressed here that the ammonia concentration measured at the surface is measured in the south of Paris in a mostly forested area that would be a sink for ammonia. Since ammonia concentration varies drastically at the surface depending on the surface itself, the surface concentration used here may not be representative of the average surface concentration of the overall region, while the concentration in the total column is representative of the overall region. Section 3.5: Consider reducing the length by being less descriptive and quantitative: Similarly to what was proposed for ammonia in a previous comment, LIDAR measurements may be used to retrieve boundary layer height that could be used to evaluate an average particle concentration, which could then be compared to the surface particles concentration. Line 567-569: Although vertical dilution is, surely an important process to explain the reduction of surface concentrations, for ammonia the surface interaction is also a key process should be considered. In particular, since surface absorption will be light dependent and temperature-dependent because of the stomatal functioning and the compensation point. See e.g. Flechard, C. R., Massad, R. S., Loubet, B., Personne, E., Simpson, D., Bash, J. O., Cooter, E. J., Nemitz, E., and Sutton, M. A.: Advances in understanding, models and parameterizations of biosphere-atmosphere ammonia exchange, Biogeosciences, 10, 5183-5225, 10.5194/bg-10-5183-2013, 2013. Sutton, M. A., Reis, S., Riddick, S. N., Dragosits, U., Nemitz, E., Theobald, M. R., Tang, Y. S., Braban, C. F., Vieno, M., Dore, A. J., Mitchell, R. F., Wanless, S., Daunt, F., Fowler, D., Blackall, T. D., Milford, C., Flechard, C. R., Loubet, B., Massad, R., Cellier, P., Personne, E., Coheur, P. F., Clarisse, L., Van Damme, M., Ngadi, Y., Clerbaux, C., Skjoth, C. A., Geels, C., Hertel, O., Kruit, R. J. W., Pinder, R. W., Bash, J. O., Walker, J. T., Simpson, D., Horvath, L., Misselbrook, T. H., Bleeker, A., Dentener, F., and de Vries, W.: Towards a climate-dependent paradigm

**ACPD**
of ammonia emission and deposition, Philos. Trans. R. Soc. B-Biol. Sci., 368, 10.1098/rstb.2013.0166, 2013. Massad, R. S., Nemitz, E., and Sutton, M. A.: Review and parameterisation of bi-directional ammonia exchange between vegetation and the atmosphere, Atmospheric Chemistry and Physics, 10, 10359-10386, 10.5194/acp-10-10359-2010, 2010.

Line 580: I might have missed something but this is the first time I see "enhanced HNO3 transport in the polluted plume". Please argue.

Figures Figure 1: please give scale on the maps. Figure 2: please explain what "a.u" means in the legend. Also please use the same units in the x-axis and in the legend text here you use 10 micrometre at once and wavenumber in cm-1. also explain what the arrows are for. Figure 3: The axes and legends are difficult to read please consider increasing the font size. Figure 4: Consider showing a graph joint with Figure 4 with the ratio of PM1 to PM 2.5 over the period. This would ease the reader to see the differences and similarities. FigurE 5: I would suggest to add a graph with the boundary layer height and another one with the average ammonia concentration in the column and considering ammonia is mainly in the boundary layer. This would allow a better comparison of the concentration at the surface and also withdraw the dilution effect due to BLH changing with time. Figure 6: To interpret Figure 6b it would be important to show the ammonia to ammonium nitrate equilibrium and its dependence on temperature and relative humidity. Figure 9: as explained before I suggest extracting the boundary layer height from these graphs and computing the average concentration over the column. This could be done in figure a and b and also for ammonia.

---

## Author Comment (AC1) · 25 May 2021

Response to comments #2

Anonymous Referee #2

This paper investigates ammonia diurnal variability near surface as well as in the tropospheric column over the megacity of Paris in spring 2012 during an intensive pollution episode and shows significant differences between them. The observations are analyzed in conjunction with particulate matter levels and meteorological parameters and the differences are explained by the dilution within the boundary layer and also by volatilization of ammonium nitrate particles. The study is interesting and contributes to the understanding of ammonia variability in the atmosphere. The manuscript is overall nicely written; however, some clarifications and corrections are needed before publication in ACP.

Response: First of all, we would like to thank again Referee#2 for the constructive and useful comments which served us as a guideline for compiling the second revision of the manuscript (RM revised manuscript hereafter). All comments are addressed as detailed below. We agree with his scientific comments and editing suggestions. We have corrected and added clarifications in the RM for addressing all of the remarks of the referee.

Line 382-384: This sentence is misleading. 'higher altitudes' needs definition because Fig 6. shows that this anti-correlation between RH and Temp is limited to the first 1000 or 1500 m.

Clarified. We have added the following clarification in Lines 409-410 of the RM: "As a conclusion, the decrease in T and the increase of RH within the boundary layer height of 1-1.5 km with respect to ground shift the equilibrium to the aerosol phase."

Line 235: Please provide further information on the model version you are using is it running off-line and if yes with what meteorological parameters. Are the CHIMERE simulations associated with ERA-Interim meteorology mentioned in lines 238-342 to be used to analyze the meteorological conditions?

Clarified and added. The paragraph in Line 248-254 was modified to provide clearer information of the use of the ESMERALDA output. "The horizontal distribution of air pollutants at the European scale is studied with CHIMERE chemistry-transport model simulations of PM2.5 provided by the ESMERALDA (EtudeS Multi RégionALes De l'Atmosphère, Cortinovis et al., 2015) project (http://www.esmeralda-web.fr/accueil/index.php). The version 2008b of CHIMERE is run hourly and averaged at daily scale, with a horizontal resolution of 15 x 15 km2 and 9 vertical levels between 20 m to 5 km. Meteorological inputs for CHIMERE come from MM5 simulations (Dudhia, 1993), using Final Analyses (FNL) data from National Centers for Environmental Prediction (NCEP) as boundary conditions."

Line 237: The reference you provide is an entire textbook. Please be specific. Which thermodynamic model is used in that simulation. (Unfortunately, the web site provided for the model in line 234 requires password, so it seems to be useless for the reader. I suggest removing it.)

Clarified and added new information. We specify the thermodynamic model input information in Lines 254-255. "Chemical reactions are simulated using the MELCHIOR2 mechanisms scheme and tabulations from ISORROPIA model for thermodynamic equilibrium calculations of the species." And changed the web site provided to one accessible without password: http://www.esmeralda-web.fr/accueil/index.php

Line 372: ISORROPIA reference is Nenes et al. 1998; ISORROPIA II reference should be Fountoukis and Nenes, ACP, 2007 please correct accordingly. Also, in which form ISORROPIA was run (forward) or backward? What input data have been used? Information on how the ISORROPIA (or ISORROPIA II) simulations are done is missing and will affect any results of the model, although in the paper such results are discussed only qualitatively.

Clarified. We use ISORROPIA II model in the forward configuration. In the RM, the reference is corrected, and we specify the type of calculation and the input species (lines 397-401) "An additional analysis was performed with the ISORROPIA II box model (Fountoukis and Nenes, 2007) to investigate the role of temperature and relative humidity in the partitioning of ammonium nitrate. The forward calculation used measurements of the SIRTA site for NH4+, NO3- and NH3 on 28 March 2012 representing the highest concentrations on the studied period, as well as the meteorological parameters. HNO3 concentrations were set constant from values in Petetin et al., (2016) for the same period of the year."

Line 354-355: figure 6a shows temperature profile up to 2.5km, the reference to the tropopause level is misleading, rephrasing is needed.

Corrected. We have rephrased the sentence only mentioning the altitudes shown in the figure (Page 10 Line 378-380 was rephrased). "During the pollution event on 26-30 March, temperature shows the usual steady decrease with altitude from the surface up to 2.5 km (see the median temperature profile measured by radiosoundings launched at Trappes on 26-30 March, Fig. 6a)."

Line 285: the discussed PM2.5 levels are the results of simulations or observations ? do in-situ observations show similar levels?

Clarified. Within the mentioned sentence the clarification that the data are simulated is provided. As well as in the following paragraph, indicating that also the in-situ measurements are similar.
Line 302-304: "On 28 March, a clear enhancement of the aerosol load over the Benelux and northern France is observed both in terms of AOD (up to 0.4) and modeled surface PM2.5 concentrations (up to 50 µg m-3)."
Line 312-313 "As polluted air masses are advected from the Benelux and west Germany on 28-29 March, PM2.5 levels clearly enhance (up to 80 µg m-3), similarly seen in daily averaged simulations."

Line 577-580: Does your model reproduce such behavior during night? The model results need to be discussed more and valorized. Does the model reproduces observed surface and profiles of NH3? The reader remains with the question why not comparing the model results to the observations?

Clarified. Due to shortcoming of total column data during the night, no detailed nocturnal evolution was considered. However, P2 implies gas-to-particle conversion at the surface, that could result in lower total column values of ammonia in the early morning (for 2 days).
Moreover, we use ISORROPIA as an equilibrium model to perform sensitivity tests. However, we would need a full 3D chemistry-transport model in order to compare modeled concentrations with observations. This is clarified in lines 625-626 of the RM. "This issue would be best addressed with chemistry-transport model simulations. "

Line 241-242: Provide temporal resolution. Are these data used as input to Chimere model?
Clarified. The temporal resolution of CHIMERE is one hour. This is clarified in line 251 of the RM.

Line 132: particulate matter
Done.

Line 211: the following
Done.

Line 297: homogeneous
Done.

Line 375: decreasing ammonia
Done.

Lines 507-508: 'depicts: : : depicting.. the last one: : :' (which one?), please rephrase the sentence.

Corrected. We have avoided the repetition of the work "depict".

---

## Author Comment (AC2) · 25 May 2021

General comments This manuscript presents ammonia total column concentration measured above Paris with an FTIR instrument (OASIS) during an intense springtime pollution episode in March 2012. The main object of the manuscript is to discuss the diel cycle of the total column NH3 concentration observed, which shows a flat profile in the morning and a steady increase in the afternoon, contrary to surface NH3 concentrations (measured at around 30 km south), that show a rather flat cycle. The manuscript provides a large overview analysis making full use of LIDAR, and surface aerosol concentration and composition, measurements as well as meteorological informations, to try explaining the total NH3 column diel cycle. The manuscript shows clearly that the surface NH3 and particulate matter concentrations are decoupled from the total column concentrations, which is a clear and important outcome of this work.

The manuscript is well structured and written and make appropriate reference to the literature (some more references are given in the detailed comments). The data presented are of great interest for the community to better understand the relationship between surface concentration and satelitte measurements for ammonia, which is a key pollutant in developped countries.

The main conclusion raised by the authors is that the observed total NH3 column diel cycle may be due to a combination of dilution (linked to the atmospheric boundary layer height cycle), aerosol volatilisation, and ammonia emissions at the regional scale. However, no clear quantitative evidence is shown to strengthen either of these conclusions, but rather qualitative arguments are given. This is the main weakness of the manuscript, which to my view, should be addressed prior to publication.

In particular, I would suggest better quantifying the effect of relative humidity and temperature on the ammonia-to-ammonium-nitrate equilibrium, and showing a graph of these dependencies. I also suggest trying to calculate, if possible, the averaged column concentration of ammonia and particles using the boundary layer height retrieved from the LIDAR measurements. I am not sure this is fully feasible, but even with some uncertainties; this may help comparing the column averaged "concentration" with the surface concentration. Additionally, the authors should bear in mind that the surface concentration of ammonia might be very much influenced by the surface surrounding the measurements, which in this case is probably a mix of forests and crops, and hence would lead to an active sink.

First of all, we would like to thank Referee#1 for the constructive and useful comments which served us as guidelines for revising our manuscript (we use RM for revised manuscript hereafter). For addressing a major suggestion of Referee#1, we had added a new figure in the paper of a comparison of vertically integrated measurements of ammonia with those at the surface multiplied by the mixing layer height, which show a good relative agreement in their diurnal evolution. We agree with his scientific comments and editing suggestions. We have corrected and added clarifications in the RM for addressing all of the remarks of the referee.

Detailed comments

Line 35: vertical dilution is not per se a variable but a process. The boundary layer height may be the variable to add here.
Agreed and corrected. We have revised this statement in Lines 34-37 as follows: "This analysis considers the following meteorological variables and processes relevant to the ammonia pollution event: temperature, relative humidity, wind speed and direction and the atmospheric boundary layer height as indicator of vertical dilution during its diurnal development."

Lines 58-59: Other effects of particle transport and deposition are acidification and loss of biodiversity (e.g. Sutton et al. 2011).
Agreed and added. We have edited Lines 59-61: "Through conversion into different forms of reactive nitrogen, further impacts of ammonia and ammonium particles are directly or indirectly linked to acidic precipitation, acidification, eutrophication and loss of biodiversity e.g. (Sutton et al., 2011; 2013; Krupa, 2003)."

Lin 59-61: This sentence on volatilisation of ammonium nitrate particles may benefit from being moved at the end of this paragraph or maybe even later in the text.
Done. This sentence was moved to the end of the paragraph (Lines 61-63).

Line 65: please provide the year of the evaluation of the total ammonia emission in Europe.
Done. We have added the year as, Line 65-66. "The main source of $NH_3$ in Europe is the agricultural sector, with an average of 93 % of total ammonia emission estimated for 2018 (Pinterits et al. 2020)."

Line 67: ammonia volatilization also depends on soil conditions like for instance humidity and pH.
Added. The authors edited the sentence in Line 66-68. "It is emitted by volatilization from fertilizer storage, livestock as well as manure and mineral nitrogen fertilizers applied on crops, as a function of temperature, humidity and pH of atmosphere and soil as well as wind speed e.g. (Sommer et al., 2004, Behera et al., 2013)."

Line 71-72: Fires are also probably a significant source of ammonia in South East Asia. Please give a range of significance of this source worldwide.
Added. We have added a sentence about source contributions, Line 74-75. "At global scale, ammonia emissions are mainly attributed to agriculture, biomass burning and the energy sector, accounting in 2005 for 80.6 %, 11.0 % and 8.3 % respectively (Behera et al., 2013)."

Line 82-83: Although it is true that agricultural activities are one of the key factor, explaining the rise in particle pollution it may be important to stress that other precursors are also essential in the process, like for instance nitrogen oxides emissions.
Added. We have added in the RM a sentence to address the raised issue, Line 83-87. "This recurrently occurs during springtime over the Paris megacity (12.2 million inhabitants including suburbs) and other European megacities often associated with emissions from agricultural activities in the areas surrounding the agglomerations e.g. Petit et al. (2015). Other pollution events in these areas are also linked to local or regional emissions of nitrogen oxides and sulfur dioxide from road traffic and industry (Behera and Sharma, 2010)."

Line 91-94: please consider reading this paper regarding stickiness of ammonia and inlet materials: Whitehead, J. D., Twigg, M., Famulari, D., Nemitz, E., Sutton, M. A., Gallagher, M. W., and Fowler, D.: Evaluation of laser absorption spectroscopic techniques for eddy covariance flux measurements of ammonia, Environmental Science & Technology, 42, 2041-2046, 2008.

Added. We have added the following information in Lines 95-100 of the RM : " Difficulties to measure ammonia by in situ techniques are associated with its "sticky" nature, inducing its accumulation in inlets and sampling tubes. In order to reduce these artefacts, different techniques are often implemented, such as the use of polyethylene or Teflon tubes (instead of steel or silicosteel), halocarbon wax coating, while keeping the length of the tubes to a minimum possible and a heating system for reducing relative humidity that may also lead to losses of $NH_3$ (Yokelson et al., 2003, Whitehead et al., 2008)."

Line 169: please explain what you mean by "co-add 30 scans".

Clarified. This term is clarified in Lines 173-176 of the RM: "The acquisition system is set to average over 30 scans at maximum spectral resolution in order to increase the signal-to-noise ratio of the measurements. This averaging procedure results in an effective temporal resolution of 10 minutes, that allows measuring the diurnal variability of relatively short-lifetime species such as $NH_3$."

Line 176-177: it would be important here better explaining how the nitric acid HNO3, in particular, is taken into account since these compound made directly interact with ammonia in the atmosphere. Although it is explained in Tournadre et al. (2020), it would be also important here to quantify the water vapour interference here.

Clarified and added. We have clarified these aspects in the following lines of the RM:
Lines 181-185: "The main interfering species in this spectral range are water vapour (H2O), carbon dioxide and O3, whose abundances are taken from the Whole Atmosphere Community Climate Model (WACCM version 6: Chang et al., 2008) and jointly adjusted with that of NH3. We also use climatological concentrations for minor interfering gases (i.e. nitric acid HNO3, sulphur hexafluoride SF6, ethane C2H4, and chlorofluorocarbons - CFC-12) that may essentially impact the baseline of the spectra."
We would like to add that PROFFIT is a radiative transfer and inversion code and thus the co-existence of several species in the atmosphere (e.g. NH3 and HNO3) is analyzed from the spectral signature of each of them.

Line 182: Retrieval error that is given between 20 and 35% may be dependent on one* or two compounds mostly. It would be good to explain the weight of water vapour and of the vertical profile shapes on this error.

Clarified. As mentioned in Tournadre et al, 2020, the total errors for the NH3 retrievals are dominated by the combination of uncertainties in the spectroscopic parameters (including also the interfering species), the noise in the spectra, the instrumental parameters, and the forward model uncertainties. They are comparable to those estimated by Dammers et al. (2015) for a high-resolution ground-based station at Bremen (Germany). Complementary to these, according to a review paper on NH3 spectroscopic parameters (Down et al., 2013), the uncertainty of 20% for NH3 line intensities used in these studies is probably a worst-case estimate and might overestimate the total errors.
We have added the following statement to clarify these aspects, Lines 188-193. "Atmospheric columns of ammonia derived from the 9 year-database of OASIS range from 0.0005 1016 to 9 1016 molecules per square centimetre (molec.cm-2) and their retrieval error is estimated to 20-35% (Tournadre et al., 2020), dominated by the systematic errors that are the combination of uncertainties in spectroscopic parameters of ammonia and the interfering species (the dominating term), radiometric noise, instrumental parameters, and forward model uncertainties. The magnitude of these errors are comparable to those estimated by Dammers et al. (2015) for a high-resolution ground-based station at Bremen (Germany)."

Line 192: It may be worth precising here that ammonia is first absorbed in acid solution.

Precision added. We have added the following statement in lines 202-205 of the RM. "The principle of this instrument, described in Cowen et al. (2004), is essentially based on conductimetric detection of ammonia that is first absorbed via a gas-permeable membrane and dissolved in water (i.e. in the form of ammonium ions, forming acidic solution). Several intercomparison exercises have shown that this procedure provides more accurate NH3 measurements (Norman et al., 2009, von Bobrutzki et al., 2010)."

Line 208: The word "pre-conditions" is unclear. Could you please rephrase?

Clarified, by editing Lines 220-222 of the RM: "The conditions needed for volatilization of NH3 from NH4NO3 are given by the relationship of relative humidity and deliquescence relative humidity (DRH), which depends on temperature."

Line 205-219: Since equilibrium between ammonia and ammonium aerosol is key in this manuscript I would suggest precising quantitatively this process. Indeed, in the current manuscript, it is said that when RH is much lower than DRH, particulate volatilization is favoured. However, volatilisation depends on the difference between RH and DRH I guess. I would therefore suggest adding a graph showing how the equilibrium between ammonia and ammonium nitrate may be dependent on relative humidity. A representation like what is shown by (for instance) Nemitz (Figure 8 of the following paper) would be interesting. Alternatively, the ISORROPIA box model could be used for that. Nemitz, E., Sutton, M. A., Wyers, G. P., Otjes, R. P., Schjoerring, J. K., Gallagher, M. W., Parrington, J., Fowler, D., and Choularton, T. W.: Surface/atmosphere exchange and chemical interaction of gases and aerosols over oilseed rape, Agric. For. Meteorol., 105, 427-445, Doi 10.1016/S0168-1923(00)00207- 0, 2000.

Agreed and clarified. We agree with the reviewer on the important role of the impact of RH on the gas-particle equilibrium of ammonium. However, it is also impacted by temperature, as well as ambient NH3 concentrations. A representation of temperature – RH dependence is already available in Wang et al. (2015) (Figure 7). It confirms that when RH is much lower than DRH, volatilization is favored. We agree that a quantitative analysis would be of high interest, but we feel that the data presented in our article is not sufficient to provide such results. We made it clear in lines 233-237 of the RM: "In addition, it is worth noting that in the present study we use the above expression and the currently available data for a qualitative interpretation of diurnal variations of ammonia and ammonium. However, additional dedicated measurements throughout the atmospheric column are needed in order to perform a quantitative analysis (see more details in the conclusion section).".

A detailed characterization of ammonia, ammonium and meteorological parameters (T, RH) should be performed throughout the column with dedicated field campaigns, offering exiting research perspectives. We mentioned that point in the conclusion, lines 636-639 of the RM: "The results of this study highlight the need of a better chemical characterization for comprehensive understanding of gas-particle partitioning over the column. A quantitative ammonia-ammonium equilibrium throughout the atmospheric column (as function of altitude) should be considered from dedicated in situ measurements field campaigns".

Line 220: I would have expected the opposite dependence of evaporation of ammonium nitrate to pH, since water solution a lower pH leads to lower hip operation. However, this may not be true in an aerosol, which is not my domain of expertise.

Clarified. As underlined, the statement applies for the pH of aerosols. The work from Guo et al., 2018 clearly suggests that formation of ammonia through volatilization is induced when the pH of the aerosols drops below 3.

Line 236: Please consider replacing "exist" by "are simulated".
Done.

Line 263: Please consider replacing "manually" by "visually".
Done.

Line 300: this is very interesting to see that PM1 is 30% lower and PM 2.5 during this episode would it be possible to propose some hypothesis ?
Clarified. The 30% difference between PM1 and PM2.5 is likely linked to aging or long-range transport as analyzed by Petit et al., (2017). We have added the following sentences in the RM (lines 322-324) "In the Paris region, the PM1 generally represent 90% of PM2.5 (Petit et al., 2017). The 30 % difference between PM1 and PM2.5 observed in the present case could be linked with aging (and/or long-range transport) which has been remarked for measurements in 2015 by Petit et al., (2017)."

Line 308: Please consider replacing "denominate these two regimes as ..." by "name these two regimes ..."
Done.

Line 320 and Figure 4: It would be worth showing the ratio of PM1 to PM2.5 in the Parisian area.
Clarified. Unfortunately, no co-located PM1 and PM2.5 are available in 2012 over the Paris area. Using measurements from different sites to calculate such ratio may be too uncertain.

Line 355: it is unclear if the Oasis inversion methodology uses this water vapour profile. Would this be of any help to better estimate water vapour interference?
Clarified. The OASIS retrieval approach uses a priori profiles for water vapor, CO2 and O3 derived from Whole Atmosphere Community Climate Model (version 6: Chang et al. 2006). These 3 species are adjusted simultaneously with NH3 for a best fit between measured and simulated radiance spectra. This is indicated in the RM (lines 181-185). For the analysis of the role of RH in ammonia/ammonium partitioning, we additionally use radiosounding measurements from Trappes (France) but they are only available at noon and midnight.

Line 372-374: it is a shame that these ISORROPIA model outputs are not shown. As already proposed in an earlier comment it would be worth showing the effect of temperature and relative humidity on the equilibrium between ammonia and ammonium nitrate.
Clarified. We agree that ammonia/ammonia partitioning as a function of relative humidity and the deliquescence point is very interesting, and it is worth dedicated analysis. However, datasets available for the current case study of March 2012 are not enough for a quantitative analysis of this topic, while ISORROPIA is not used for a direct comparison with measurements as it is not a full 3D model. Future studies will address these aspects, including 3D chemistry-transport modelling and dedicated field campaigns in situ measurements throughout the atmospheric column.
As mentioned for the remarks of Lines 205-219, we clarified these aspects in lines 233-237 and 636-639 of the RM (see above comment).

Line 377: the meaning of the sentence "vertically decreasing variation of ammonia with increasing altitudes" is unclear. Please rephrase.
Done. The sentence was rephrased in the RM as follows (line 403-405). "… ammonia likely decreases at higher altitudes."

Line 420: It may be worth looking at the following paper that reported ammonia emissions and concentrations during the very same period in the west of Paris: Personne, E., Tardy, F., Genermont, S., Decuq, C., Gueudet, J. C., MASCHER, N., Durand, B., Masson, S., Lauransot, M., Flechard, C., Burkhardt, J., and Loubet, B.: Investigating sources and sinks for ammonia exchanges between the atmosphere and a wheat canopy following slurry application with trailing hose, Agric. For. Meteorol., 207, 11-23, 10.1016/j.agrformet.2015.03.002, 2015.

Cited and added additional information. We have added the following information from the suggested reference. Lines 446-448 "Additionally, volatilization of applied mineral fertilizers in the surrounding crop areas may also contribute to the daytime increase of ammonia, as analyzed in detail during the same period (March/April 2012) over crop fields located west of Paris by Personne et al. (2015)."

Line 471-475 "Ammonia emissions from the soil of surrounding crop areas may contribute to its enhancement during the morning (Personne et al., 2015)."

Lines 422-249: The manuscript would benefit a lot if some more quantitative estimation of the effects of the boundary layer height on the concentration of the total ammonia column concentration could be estimated even roughly. Would it be feasible to use the atmospheric boundary layer height as retrieved from the LIDAR measurements shown in figure 9 to estimate an averaged ammonia concentration in the column?

Done with an additional figure (new Figure 10). We appreciate this very useful suggestion and thank the referee for it. We have followed this suggestion and compared total column retrievals of ammonia with surface measurements multiplied by the atmospheric mixing boundary layer derived by the lidar. The comparison shows a very good relative agreement in the diurnal evolution of these amounts and confirms the major role of vertical mixing for comparing these complementary data. This match in relative terms is found both for periods P1 and P2. Differences in absolute terms likely come from the evolution of the vertical profile of ammonia and also the variability of ammonia abundance in the residual boundary layer and the free troposphere above.

These comments are added in the RM as (lines 561-571): "An additional analysis that highlights the major role of vertical mixing for comparing vertically integrated and surface measurements of ammonia is shown in Figure 10. For both periods, we compare the daily evolution of total column of NH3 retrieved by OASIS with that of surface measurements of ammonia multiplied by the atmospheric mixing boundary layer derived from lidar measurements (magenta lines in Fig. 9c-d). This last one corresponds to the vertically integrated amount of ammonia over the mixing layer for the case of a vertically homogenous distribution of this gas. We clearly remark a very similar diurnal evolution of these two quantities in relative terms, for both periods. This confirms the good consistency between these two independent measurements of ammonia (total column and surface data). Differences in absolute terms (between 0.5 to 1 $10^6$ molec. cm-2) likely come from the evolution of the vertical profile of ammonia, changes with respect to the vertically homogenous distribution, and also the variability of ammonia abundance in the residual boundary layer and the free troposphere above the mixing layer."

Lines 440-445: it should be stressed here that the ammonia concentration measured at the surface is measured in the south of Paris in a mostly forested area that would be a sink for ammonia. Since ammonia concentration varies drastically at the surface depending on the surface itself, the surface concentration used here may not be representative of the average surface concentration of the overall region, while the concentration in the total column is representative of the overall region.

Done and clarified. Following the referee's suggestion, we have added the following statement in lines 473-475. "It is also worth noting that forests surrounding the NH3 surface measurement site at Palaiseau

may act local sinks of ammonia (as remarked by Behera et al., 2013, Hansen et al., 2015), this is not the case for total column retrievals performed at Créteil."

Section 3.5: Consider reducing the length by being less descriptive and quantitative: Similarly to what was proposed for ammonia in a previous comment, LIDAR measurements may be used to retrieve boundary layer height that could be used to evaluate an average particle concentration, which could then be compared to the surface particles concentration.

Clarified. Although this section could be reduced for simplicity, we think that it is important to clearly describe each of the multiple elements considered in the analysis: ammonia, particles, their interactions, vertical distribution, surface concentrations and vertically integrated amounts. A relatively original aspect of this analysis is to use particle abundance tracers (derived directly from lidar) as proxy of ammonia vertical distribution. Then in a further step we compare total integrated concentrations of ammonia and those estimated from surface data and the boundary layer height (although here we have significant ambiguities with respect to the true vertical profile of ammonia which has not been measured for our analysis). Therefore, we believe it is important to explain and reinforce our conclusions in a multiple perspective: comparing direct measurements themselves and also combining them to derive vertically integrated estimates in both cases.

Line 567-569: Although vertical dilution is, surely an important process to explain the reduction of surface concentrations, for ammonia the surface interaction is also a key process should be considered. In particular, since surface absorption will be light dependent and temperature-dependent because of the stomatal functioning and the compensation point. See e.g.

Flechard, C. R., Massad, R. S., Loubet, B., Personne, E., Simpson, D., Bash, J. O., Cooter, E. J., Nemitz, E., and Sutton, M. A.: Advances in understanding, models and parameterizations of biosphere-atmosphere ammonia exchange, Biogeosciences, 10, 5183-5225, 10.5194/bg-10-5183-2013, 2013.

Sutton, M. A., Reis, S., Riddick, S. N., Dragosits, U., Nemitz, E., Theobald, M. R., Tang, Y. S., Braban, C. F., Vieno, M., Dore, A. J., Mitchell, R. F., Wanless, S., Daunt, F., Fowler, D., Blackall, T. D., Milford, C., Flechard, C. R., Loubet, B., Massad, R., Cellier, P., Personne, E., Coheur, P. F., Clarisse, L., Van Damme, M., Ngadi, Y., Clerbaux, C., Skjoth, C. A., Geels, C., Hertel, O., Kruit, R. J. W., Pinder, R. W., Bash, J. O., Walker, J. T., Simpson, D., Horvath, L., Misselbrook, T. H., Bleeker, A., Dentener, F., and de Vries, W.: Towards a climate-dependent paradigm of ammonia mission and deposition, Philos. Trans. R. Soc. B-Biol. Sci., 368, 10.1098/rstb.2013.0166, 2013.

Massad, R. S., Nemitz, E., and Sutton, M. A.: Review and parameterisation of bi-directional ammonia exchange between vegetation and the atmosphere, Atmospheric Chemistry and Physics, 10, 10359-10386, 10.5194/acp-10-10359-2010, 2010.

Agreed and added. We have added an additional statement to clarified this aspect in the conclusions (lines 611-613): "Other processes such as surface and canopy uptake from surrounding ecosystems, depending on pH and total nitrogen input, may also explain surface concentration reductions (Massad et al., 2010, Flechard et al., 2013, Personne et al., 2015)."

Line 580: I might have missed something but this is the first time I see "enhanced HNO3 transport in the polluted plume". Please argue.

Suppressed. The possible enhancement of HNO3 is just a hypothesis here and therefore we have suppressed the statement for the RM.

Figures Figure 1: please give scale on the maps.
Done.

Figure 2: please explain what "a.u" means in the legend. Also please use the same units in the x-axis and in the legend text here you use 10 micrometre at once and wavenumber in cm-1. also explain what the arrows are for.

Clarified. a.u. means arbitrary unit. The authors modified the legend of Fig 2 by adding the origin of the ν2 band of ammonia in wavenumber (932 cm-1) and added "All spectra are plotted using arbitrary unit (a.u.) on the Y scale".

Figure 3: The axes and legends are difficult to read please consider increasing the font size.
Done.

Figure 4: Consider showing a graph joint with Figure 4 with the ratio of PM1 to PM 2.5 over the period. This would ease the reader to see the differences and similarities.
Clarified. Since no co-localised measurements of PM1 and PM2.5 are available, we prefer not to calculate the ratio of these amounts measured at different locations and it would be too uncertain.

Figure 5: I would suggest to add a graph with the boundary layer height and another one with the average ammonia concentration in the column and considering ammonia is mainly in the boundary layer. This would allow a better comparison of the concentration at the surface and also withdraw the dilution effect due to BLH changing with time.
Done. This new analysis is done in the new Figure 10.

Figure 6: To interpret Figure 6b it would be important to show the ammonia to ammonium nitrate equilibrium and its dependence on temperature and relative humidity.
Clarified. The suggestion is interesting; however, the available data is not enough to perform a quantitative analysis. This aspect will be addressed in future studies (see answer to comments on Line 209-215 for further details)

Figure 9: as explained before I suggest extracting the boundary layer height from these graphs and computing the average concentration over the column. This could be done in figure a and b and also for ammonia.
Done. This new analysis is done in the new Figure 10.

---

## Author Response (AR2)

**Anonymous Referee #2 - Comments on REVISED SUBMISSION**

The authors have addressed all comments by both reviewers and the presentation of the results and the discussion have been improved. To my opinion, a few additional improvements are needed before publication of the paper in ACP.

1- Abstract: line 44-46 :"These differences are mainly explained by vertical mixing within the boundary layer, as suggested by ground-based measurements of vertical profiles of aerosol backscatter, used as tracer of the vertical distribution of pollutants in the atmospheric boundary layer"

New Comment: From your new analysis (new figure 10), i.e. calculation of the boundary layer column of NH3 as suggested by reviewer Benjamin Loubet, I understand that the difference in the diurnal patterns between surface concentrations and total column measurements can be mostly attributed to the changes in the mixing layer height, provided that boundary layer is considered well mixed and therefore homogeneous in ammonia concentrations (new Figure 10). This finding has to reflect in the abstract.

Done. We have added this aspect in lines 45-46 of the revised manuscript: "*These differences are mainly explained by vertical mixing within the boundary layer, provided that this last one is considered well mixed and therefore homogeneous in ammonia concentrations. This is suggested by ground-based measurements of vertical profiles of aerosol backscatter, used as tracer of the vertical distribution of pollutants.*"

2- Earlier reviewer comment: Line 237: The reference you provide is an entire textbook. Please be specific. Which thermodynamic model is used in that simulation. (Unfortunately, the web site provided for the model in line 234 requires password, so it seems to be useless for the reader. I suggest removing it.)
Author reply: Clarified and added new information. We specify the thermodynamic model input information in Lines 254-255. "Chemical reactions are simulated using the MELCHIOR2 mechanisms scheme and tabulations from ISORROPIA model for thermodynamic equilibrium calculations of the species." And changed the web site provided to one accessible without password: http://www.esmeralda-web.fr/accueil/index.php

New comment: I am afraid it remains unclear to me: Do I understand correct that ISORROPIA model has been used to produce tables that have been then inserted in the CHIMERE model? Or is only that the equilibrium coefficients and other thermodynamic input data from ISORROPIA have been used in some module of Chimere to calculate the equilibrium? Please further clarify.

Clarified. ISORROPIA is used to produce tables that are inserted in the chemistry-transport model for calculating the thermodynamic equilibrium of the species.

This is stated in the manuscript as (lines 254-257): *"Chemical reactions are simulated using the MELCHIOR2 mechanisms scheme and the ISORROPIA model (Nenes et al., 1998). This last one has been used to produce tables that are inserted in the CHIMERE model for calculating the thermodynamic equilibrium of the species."*

3- New Comment: Line 313: PM2.5 levels ARE clearly enhancED (up to 80 ug m-3), as also seen in the daily averageD results of the simulations.

Done.

4- New Comment: Line 323: Reading the manuscript I understand that PM1 was measured at SIRTA while PM2.5 at other location in the Ile-de-France region. Thus, PM1 and PM2.5 measurements are not co-located. These two types of measurements could be compared when the measurements are co-located, which -to my understanding - is not the case, or when the PM2.5 measurements could be considered as characteristic of the region. It is true that the 3 PM2.5 sites provide very similar results, with the exception of the night of the 26-27 March but SIRTA is not included in the triangle defined by the PM2.5 sites. I do not see how the authors exclude the fact that there might be differences in the background levels at SIRTA and the other stations as well as different sources and sinks. Do the authors have another argument/evidence that the 3 PM2.5 stations are representative of the entire area including SIRTA? As mentioned only with co-located observations one can relate PM1 and PM2.5 observation.

Clarified and agreed. These two aspects are clarified in the following:

The SIRTA site measurements of PM1 are statistically consistent over a long period with respect to PM2.5 data from Gennevilliers, Bobigny and Vitry-sur-Seine. This is shown by the ancillary figure 17 from Petit (2014) shown below. We observe that the statistical distribution over 2 years of the measurements of PM2.5 at the urban background sites Gennevilliers, Bobigny and Vitry-sur-Seine of AIRPARIF (the ones from Figure 4 of the manuscript Kutzner et al.) are very similar to that of PM1 at the SIRTA site. This is clearly not the case of traffic (A1, AUT, BP_EST) nor rural stations (RUR_S).

We agree that PM1 measurements at SIRTA might occasionally differ in terms of background levels with respect to PM2.5 at this site itself and with respect to the other 3 sites. However, most (larger than 80 %) of the variability of PM2.5 during pollution events are measured at SIRTA in terms of PM1. This is illustrated in the Figure 18 from Petit (2014) showing the difference between PM1 and PM2.5 at the SIRTA site itself for one month in 2013. We clearly remark that the largest variability is observed for the PM1 fraction, which represents in average 85% of PM2.5, over the period of comparison. In cases of relatively higher pollution level (PM1 larger than 20 μg/m3), PM1 is larger than 80% of PM2.5 while in is about 50% for low pollution levels.

These aspects are clarified in the revised manuscript as (lines 325-329) "*In the Paris region, the PM1 generally represent 90% of PM2.5 (Petit et al., 2017), particularly when PM1 is larger than 20 μg m-3 (although for lower levels, PM1 may represent around 50% of PM2.5, Petit (2014)). Occasionally, some background levels of PM might not be accounted in PM1 that are measured as PM2.5 (Petit, 2014). Moreover, comparisons made by Petit (2014) show a very similar statistical distribution for both PM1 at SIRTA and PM2.5 at the urban background stations at Paris suburbs mentioned in Fig. 4.*"

[Figure]

Figure 17 of Petit (2014): Variability in terms of box plots of PM observations from 01/04/2012 to 01/05/2014 from traffic stations (A1, AUT, BP_EST), urban background station (PA04C Paris downtown, GEN: Gennevilliers, BOB: Bobigny and VITRY: Vitry sur Seine), a rural station (Bois-Herpin) and the SIRTA site (only this one is PM1 while others are PM2.5).

[Figure]

Figure 18 of Petit (2014): (Left) time evolution of PM1 (diameter smaller than 1 µm) and PM2.5-PM1 (diameter between 1 and 2.5 µm from November to December 2013 measured at SIRTA. (Right) scatterplot of the ratio PM1/PM2.5 with respect to PM1 (44032 minute-resolution points)

Petit, J.-E. : Compréhension des sources et des processus de formation de la pollution particulaire en région Ile-de-France, PhD manuscript of the University of Versailles Saint-Quentin-en-Yvelines, France, 2014

5- Line 473: act AS local sinks

Done.

6- Author reply: This is clarified in lines 625-626 of the RM. "This issue would be best addressed with chemistry-transport model simulations. "

New Comment: This sentence is misleading because ISORROPIA II is a thermodynamic equilibrium model and CHIMERE is a chemistry transport model. What is needed is coupling the CTM with ISORROPIA, and the reader is wondering why these two have not been coupled for this study to fully address the gas-to-particle conversion issue of ammonia. In addition, CHIMERE model output could have been easily used off line in the ISORROPIA II model. A better argument is needed here.

Clarified. It is true that ISORROPIA II is a module integrated (as tables in the version used in the paper) within the chemistry transport model CHIMERE. However, we would need to parametrize and validate it to use it to describe particle-gas partitioning for the specific case analyzed in the paper and the region of the Paris megacity. The current low accuracy for simulating ammonia/ammonium nitrite gas-to-particle conversion with a chemistry transport model is illustrated by Petetin et al. (2016). A dedicated study of parametrization and validation of the model would need co-located in situ measurements of particulate nitrate, ammonia, and nitric acid. Therefore, such study is out the scope of the present paper.

We have added the following comment to clarify this aspect (lines 629-631) "*This issue would be best addressed with chemistry-transport model simulations and dedicated in situ measurements (including nitric acid, particulate nitrate and ammonia) for the parametrization and validation of the model. "*

7- Author reply lines 611-614: Other processes such as surface and canopy uptake from surrounding ecosystems, depending on pH and total nitrogen input, may also explain surface concentration reductions (Massad et al., 2010, Flechard et al., 2013, Personne et al., 2015).

New Comment: Please also add light and temperature.

Done.